# Can contrastive learning avoid shortcut solutions?

**Joshua Robinson**
MIT CSAIL & LIDS
joshrob@mit.edu

**Li Sun**
University of Pittsburgh
lis118@pitt.edu

**Ke Yu**
University of Pittsburgh
yu.ke@pitt.edu

**Kayhan Batmanghelich**
University of Pittsburgh
kayhan@pitt.edu

**Stefanie Jegelka**
MIT CSAIL
stefje@csail.mit.edu

**Suvrit Sra**
MIT LIDS
suvrit@mit.edu

## Abstract

The generalization of representations learned via contrastive learning depends crucially on what features of the data are extracted. However, we observe that the contrastive loss does not always sufficiently guide which features are extracted, a behavior that can negatively impact the performance on downstream tasks via "shortcuts", i.e., by inadvertently suppressing important predictive features. We find that feature extraction is influenced by the *difficulty* of the so-called instance discrimination task (i.e., the task of discriminating pairs of similar points from pairs of dissimilar ones). Although harder pairs improve the representation of some features, the improvement comes at the cost of suppressing previously well represented features. In response, we propose *implicit feature modification* (IFM), a method for altering positive and negative samples in order to guide contrastive models towards capturing a wider variety of predictive features. Empirically, we observe that IFM reduces feature suppression, and as a result improves performance on vision and medical imaging tasks. The code is available at: https://github.com/joshr17/IFM.

## 1 Introduction

Representations trained with contrastive learning are adept at solving various vision tasks including classification, object detection, instance segmentation, and more [5, 15, 44]. In contrastive learning, encoders are trained to discriminate pairs of positive (similar) inputs from a selection of negative (dissimilar) pairs. This task is called *instance discrimination*: It is often framed using the InfoNCE loss [14, 33], whose minimization forces encoders to extract input features that are sufficient to discriminate similar and dissimilar pairs.

However, learning features that are discriminative during training does not guarantee a model will generalize. Many studies find inductive biases in supervised learning toward *simple* "shortcut" features and decision rules [16, 21, 32] which result in unpredictable model behavior under perturbations [22, 43] and failure outside the training distribution [2, 37]. Simplicity bias has various potential sources [11] including training methods [8, 29, 41] and architecture design [10, 17]. Bias towards shortcut decision rules also hampers transferability in contrastive learning [4], where it is in addition influenced by the instance discrimination task. These difficulties lead us to ask: can the contrastive instance discrimination task itself be modified to avoid learning shortcut solutions?

We approach this question by studying the relation between contrastive instance discrimination and feature learning. First, we theoretically explain why optimizing the InfoNCE loss alone does not guarantee avoidance of shortcut solutions that *suppress* (i.e., discard) certain input features [4, 11]. Second, despite this negative result, we show that it is still possible to trade off representation of

---

Correspondence to Joshua Robinson (joshrob@mit.edu).

35th Conference on Neural Information Processing Systems (NeurIPS 2021).

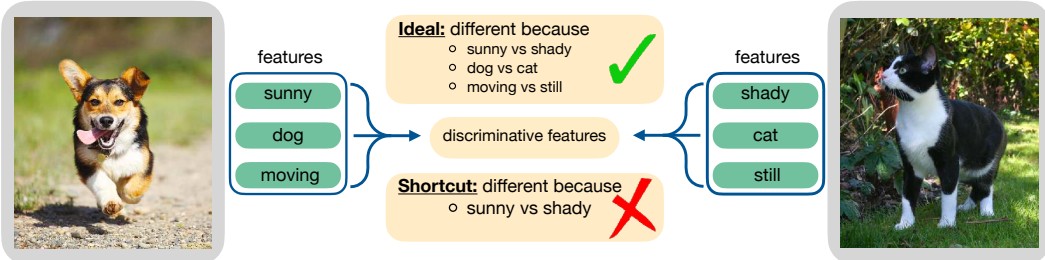

Figure 1: An ideal encoder would discriminate between instances using multiple distinguishing features instead of finding simple shortcuts that suppress features. We show that InfoNCE-trained encoders can suppress features (Sec. 2.2). However, making instance discrimination harder during training can trade off representation of different features (Sec. 2.3). To avoid the need for trade-offs we propose *implicit feature modification* (Sec. 3), which reduces suppression in general, and improves generalization (Sec. 4).

one feature for another using simple methods for adjusting the difficulty of instance discrimination. However, these methods have an important drawback: improved learning of one feature often comes at the cost of harming another. That is, feature suppression is still prevalent. In response, we propose *implicit feature modification*, a technique that encourages encoders to discriminate instances using multiple input features. Our method introduces no computational overhead, reduces feature suppression (without trade-offs), and improves generalization on various downstream tasks.

**Contributions.** In summary, this paper makes the following main contributions:

1. It analyzes feature suppression in contrastive learning, and explains why feature suppression can occur when optimizing the InfoNCE loss.

2. It studies the relation between instance discrimination tasks and feature learning; concretely, adjustments to instance discrimination difficulty leads to different features being learned.

3. It proposes *implicit feature modification*, a simple and efficient method that reduces the tendency to use feature suppressing shortcut solutions and improves generalization.

## 1.1 Related work

Unsupervised representation learning is enjoying a renaissance driven by steady advances in effective frameworks [3, 5, 15, 18, 33, 44, 45, 51]. As well as many effective contrastive methods, Siamese approaches that avoid representation collapse without explicitly use of negatives have also been proposed [6, 13, 51]. Pretext task design has been at the core of progress in self-supervised learning. Previously popular tasks include image colorization [54] and inpainting [35], and theoretical work shows pre-trained encoders can provably generalize if a pretext task necessitates the learning of features that solve downstream tasks [27, 39]. In contrastive learning, augmentation strategies are a key design component [5, 48, 50], as are negative mining techniques [9, 15, 25, 40]. While feature learning in contrastive learning has received less attention, recent work finds that low- and mid-level features are more important for transfer learning [55], and feature suppression can occur [4] just as with supervised learning [10, 16]. Combining contrastive learning with an auto-encoder has also been considered [28], but was found to harm representation of some features in order to avoid suppression of others. Our work is distinguished from prior work through our focus on how the design of the instance discrimination task itself affects which features are learned.

## 2 Feature suppression in contrastive learning

Feature suppression refers to the phenomenon where, in the presence of multiple predictive input features, a model uses only a subset of them and ignores the others. The selected subset often corresponds to intuitively "simpler" features, e.g., color as opposed to shape. Such features lead to "shortcut" decision rules that might perform well on training data, but can harm generalization and lead to poor robustness to data shifts. Feature suppression has been identified as a common problem in deep learning [11], and both supervised and contrastive learning suffer from biases induced by the choice of optimizer and architecture. However, contrastive learning bears an additional potential source of bias: *the choice of instance discrimination task*. Which positive and negative pairs are presented critically affects which features are discriminative, and hence which features are learned. In this work we study the relation between feature suppression and instance discrimination.

First, we explain why optimizing the InfoNCE loss is insufficient in general to avoid feature suppression, and show how it can lead to counter-intuitive generalization (Sec. 2.2). Given this negative result, we then ask if it is at least possible to *control* which features a contrastive encoder learns? We find that this is indeed the case, and that adjustments to the instance discrimination task lead to different features being learned (Sec. 2.3). However, the primary drawback of these adjustments is that improving one feature often comes at the cost of harming representation of another. That is, feature suppression is still prevalent. Addressing this drawback is the focus of Sec. 3.

## 2.1 Setup and definition of feature suppression

Formally, we assume that the data has underlying feature spaces $\mathcal{Z}^1, \ldots, \mathcal{Z}^n$ with a distribution $p_j$ on each $\mathcal{Z}^j$. Each $j \in [n]$, corresponding to a latent space $\mathcal{Z}^j$, models a distinct feature. We write the product as $\mathcal{Z}^S = \prod_{j \in S} \mathcal{Z}^j$, and simply write $\mathcal{Z}$ instead of $\mathcal{Z}^{[n]}$ where $[n] = \{1, \ldots, n\}$. A set of features $z = (z^j)_{j \in [n]} \in \mathcal{Z}$ is generated by sampling each coordinate $z^j \in \mathcal{Z}^j$ independently, and we denote the measure on $\mathcal{Z}$ induced by $z$ by $\lambda$. Further, let $\lambda(\cdot | z^S)$ denote the conditional measure on $\mathcal{Z}$ for fixed $z^S$. For $S \subseteq [n]$ we use $z^S$ to denote the projection of $z$ onto $\mathcal{Z}^S$. Finally, an injective map $g : \mathcal{Z} \to \mathcal{X}$ produces observations $x = g(z)$.

Our aim is to train an encoder $f : \mathcal{X} \to \mathbb{S}^{d-1}$ to map input data $x$ to the surface of the unit sphere $\mathbb{S}^{d-1} = \{u \in \mathbb{R}^d : \|u\|_2 = 1\}$ in such a way that $f$ extracts useful information. To formally define feature suppression, we need the *pushforward* $h \# \nu(V) = \nu(h^{-1}(V))$ of a measure $\nu$ on a space $\mathcal{U}$ for a measurable map $h : \mathcal{U} \to \mathcal{V}$ and measurable $V \subseteq \mathcal{V}$, where $h^{-1}(V)$ denotes the preimage.

Consider an encoder $f : \mathcal{X} \to \mathbb{S}^{d-1}$ and features $S \subseteq [n]$. For each $z^S \in \mathcal{Z}^S$, let $\mu(\cdot | z^S) = (f \circ g) \# \lambda(\cdot | z^S)$ be the pushforward measure on $\mathbb{S}^{d-1}$ by $f \circ g$ of the conditional $\lambda(\cdot | z^S)$.

1. $f$ suppresses $S$ if for any pair $z^S, \bar{z}^S \in \mathcal{Z}^S$, we have $\mu(\cdot | z^S) = \mu(\cdot | \bar{z}^S)$.
2. $f$ distinguishes $S$ if for any pair of distinct $z^S, \bar{z}^S \in \mathcal{Z}^S$, measures $\mu(\cdot | z^S), \mu(\cdot | \bar{z}^S)$ have disjoint support.

Feature suppression is thus captured in a distributional manner, stating that $S$ is suppressed if the encoder distributes inputs in a way that is invariant to the value $z^S$. Distinguishing features, meanwhile, asks that the encoder $f$ separates points with different features $z^S$ into disjoint regions. We consider training an encoder $f : \mathcal{X} \to \mathbb{S}^{d-1}$ to optimize the InfoNCE loss [33, 14],

$$\mathcal{L}_m(f) = \mathbb{E}_{x, x^+, \{x_i^-\}_{i=1}^m} \left[ -\log \frac{e^{f(x)^\top f(x^+)/\tau}}{e^{f(x)^\top f(x^+)/\tau} + \sum_{i=1}^m e^{f(x)^\top f(x_i^-)/\tau}} \right], \tag{1}$$

where $\tau$ is known as the *temperature*. Positive pairs $x, x^+$ are generated by first sampling $z \sim \lambda$, then independently sampling two random augmentations $a, a^+ \sim \mathcal{A}$, $a : \mathcal{X} \to \mathcal{X}$ from a distribution $\mathcal{A}$, and setting $x = a(g(z))$ and $x^+ = a^+(g(z))$. We assume $\mathcal{A}$ samples the identity function $a(x) = x$ with non-zero probability ("$x$ is similar to itself"), and that there are no collisions: $a(x) \neq a'(x')$ for all $a, a'$, and all $x \neq x'$. Each negative example $x_i^-$ is generated as $x_i^- = a_i(g(z_i))$, by independently sampling features $z_i \sim \lambda$ and an augmentation $a_i \sim \mathcal{A}$.

## 2.2 Why optimizing the InfoNCE loss can still lead to feature suppression

Do optimal solutions to the InfoNCE loss automatically avoid shortcut solutions? Unfortunately, as we show in this section, this is not the case in general; there exist both optimal solutions of the InfoNCE loss that do and solutions that do not suppress a given feature. Following previous work [40, 49, 56], we analyze the loss as the number of negatives goes to infinity,

$$\mathcal{L} = \lim_{m \to \infty} \left\{ \mathcal{L}_m(f) - \log m - \frac{2}{\tau} \right\} = \frac{1}{2\tau} \mathbb{E}_{x, x^+} \|f(x) - f(x^+)\|^2 + \mathbb{E}_{x^+} \log \left[ \mathbb{E}_{x^-} e^{f(x^+)^\top f(x^-)/\tau} \right].$$

We subtract $\log m$ to ensure the limit is finite, and use $x^-$ to denote a random sample with the same distribution as $x_i^-$. Prop. 2.2 (proved in App. A) shows that, assuming the marginals $p_j$ are uniform, the InfoNCE loss is optimized both by encoders that suppress feature $j$, and by encoders that distinguish $j$.

Suppose that $p_j$ is uniform on $\mathcal{Z}^j = \mathbb{S}^{d-1}$ for all $j \in [n]$. Then for any feature $j \in [n]$ there exists an encoder $f_{\text{supp}}$ that suppresses feature $j$ and encoder $f_{\text{disc}}$ that discriminates $j$ but both attain $\min_{f : \text{measurable}} \mathcal{L}(f)$.

The condition that $p_j$ is uniformly distributed on $\mathcal{Z}^j = \mathbb{S}^{d-1}$ is similar to conditions used in previous work [56]. Prop. 2.2 shows that empirical observations of feature suppression [4] (see also Fig. 3)

are not simply due to a failure to sufficiently optimize the loss, but that the possibility of feature suppression is *built into* the loss. What does Prop. 2.2 imply for the generalization behavior of encoders? Besides explaining why feature suppression can occur, Prop. 2.2 also suggests another counter-intuitive possibility: *lower InfoNCE loss may actually lead to worse performance on some tasks*.

To empirically study whether this possibility manifests in practice, we use two datasets with known semantic features: (1) In the Trifeature data, [16] each image is $128 \times 128$ and has three features: color, shape, and texture, each taking possible 10 values. See Fig. 10, App. C for sample images. (2) In the STL-digits data, samples combine MNIST digits and STL10 objects by placing copies of a randomly selected MNIST digit on top of an STL10 image. See Fig. 11 App. C for sample images.

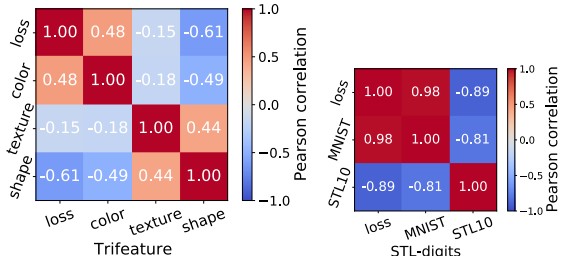

Figure 2: Linear readout error on different downstream tasks can be negatively correlated. Further, lower InfoNCE loss does not always yield not lower error: error rates on texture, shape and STL10 prediction are *negatively correlated* with InfoNCE loss.

We train encoders with ResNet-18 backbone using SimCLR [5]. To study correlations between the loss value and error on downstream tasks, we train 33 encoders on Trifeature and 7 encoders on STL-digits with different hyperparameter settings (see App. C.2 for full details on training and hyperparameters). For Trifeature, we compute the Pearson correlation between InfoNCE loss and linear readout error when predicting {color, shape, texture}. Likewise, for STL-digits we compute correlations between the InfoNCE loss and MNIST and STL10 prediction error.

Fig. 2 shows that performance on different downstream tasks is not always positively correlated. For Trifeature, color error is negatively correlated with shape and texture, while for STL-digits there is a strong negative correlation between MNIST digit error and STL10 error. Importantly, lower InfoNCE loss is correlated with lower prediction error for color and MNIST-digit, but with *larger* error for shape, texture and STL10. Hence, lower InfoNCE loss can improve representation of some features (color, MNIST digit), but may actually *hurt* others. This conflict is likely due to the simpler color and MNIST digit features being used as shortcuts. Our observation is an important addition to the statement of Wang and Isola [49] that lower InfoNCE loss improves generalization: the situation is more subtle – whether lower InfoNCE helps generalization on a task depends on the use of shortcuts.

## 2.3 Controlling feature learning via the difficulty of instance discrimination

The previous section showed that the InfoNCE objective has solutions that suppress features. Next, we ask what factors determine which features are suppressed? Is there a way to target *specific* features and ensure they are encoded? One idea is to use *harder* positive and negative examples. Hard examples are precisely those that are not easily distinguishable using the currently extracted features. So, a focus on hard examples may change the scope of the captured features. To test this hypothesis, we consider two methods for adjusting the difficulty of positive and negative samples:

1. Temperature $\tau$ in the InfoNCE loss (Eqn. 1). Smaller $\tau$ places higher importance on positive an negative pairs with high similarity [47].
2. Hard negative sampling method of Robinson et al. [40], which uses importance sampling to sample harder negatives. The method introduces a hardness concentration parameter $\beta$, with larger $\beta$ corresponding to harder negatives (see [40] for full details).

Results reported in Fig. 3 (also Fig. 13 in App. C.2) show that varying instance discrimination difficulty—i.e., varying temperature $\tau$ or hardness concentration $\beta$—enables trade-offs between which features are represented. On Trifeature, easier instance discrimination (large $\tau$, small $\beta$) yields good performance on 'color'—an "easy" feature for which a randomly initialized encoder already has high linear readout accuracy—while generalization on the harder texture and shape features is poor. The situation *reverses* for harder instance discrimination (small $\tau$, large $\beta$). We hypothesize that the use of "easy" features with easy instance discrimination is analogous to simplicity biases in supervised deep networks [17, 21]. As with supervised learning [10, 17], we observe a bias for texture over shape in convolutional networks, with texture prediction always outperforming shape.

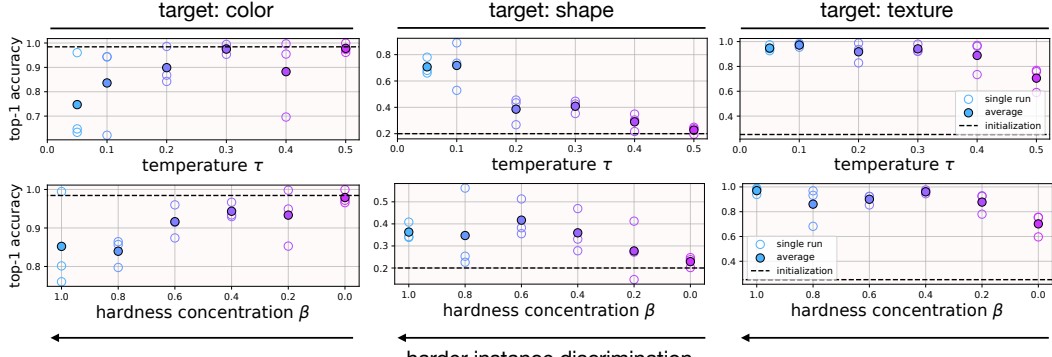

Figure 3: Trifeature dataset [16]. The *difficulty* of instance discrimination affects which features are learned (Sec. 2.3). When instance discrimination is easy (big $\tau$, small $\beta$), encoders represent color well and other features badly. When instance discrimination is hard (small $\tau$, big $\beta$), encoders represent more challenging shape and texture features well, at the expense of color.

That there are simple levers for controlling which features are learned already distinguishes contrastive learning from supervised learning, where attaining such control is less easy (though efforts have been made [23]). However, these results show that representation of one feature must be sacrificed in exchange for learning another one better. To understand how to develop methods for improving feature representation without suppressing others, the next result (proof in App. A) examines more closely *why* there is a relationship between (hard) instance discrimination tasks and feature learning.

[Informal] Suppose that $p_j$ is uniform on $\mathcal{Z}^j = \mathbb{S}^{d-1}$ for all $j \in [n]$. Further, for $S \subseteq [n]$ suppose that $x, x^+, \{x_i^-\}_i$ are conditioned on the event that they have the same features $S$. Then any $f$ that minimizes the (limiting) InfoNCE loss suppresses features $S$.

The positive and negative instances in Prop. 2.3 must be distinguished with features in $S^c$. Relating this point to the above observations, assume that an encoder exclusively uses features $S$. Any positives and negatives that do not (much) differ in features $S$ are difficult for the encoder. By Prop. 2.3, focusing the training on these difficult examples pushes the encoder to instead use features in $S^c$, i.e., to learn *new* features. But at the same time, the proposition also says that a strong focus on such hard negative pairs leads to suppressing the originally used features $S$, explaining the results in Fig. 3. While the two techniques for adjusting instance difficulty we studied were unable to avoid feature suppression, this insight forms the motivation for *implicit feature modification*, which we introduce next.

## 3 Implicit feature modification for reducing feature suppression

The previous section found that simple adjustments to instance discrimination *difficulty* could significantly alter which features a model learns. Prop. 2.3 suggests that this ability to modify which features are learned stems from holding features constant across positive and negative samples. However, these methods were unable to avoid trade-offs in feature representation (Fig. 3) since features that are held constant are themselves suppressed (Prop. 2.3).

To avoid this effect, we develop a technique that *adaptively* modifies samples to remove whichever features are used to discriminate a particular positive pair from negatives, then trains an encoder to discriminate instances using *both* the original features, and the features left over after modification. While a natural method for modifying features is to directly transform raw input data, it is very challenging to modify the semantics of an input in this way. So instead we propose modifying features by applying transformations to encoded samples $v = f(x)$. Since we modify the encoded samples, instead of raw inputs $x$, we describe our method as *implicit*.

We set up our notation. Given batch $x, x^+, \{x_i^-\}_{i=1}^m$ we write $v = f(x)$, $v^+ = f(x^+)$, and $v_i^- = f(x_i^-)$ to denote the corresponding embeddings. As in Eqn. 1, the point-wise InfoNCE loss is,

$$\ell(v, v^+, \{v_i^-\}_{i=1}^m) = -\log \frac{e^{v^\top v^+/\tau}}{e^{v^\top v^+/\tau} + \sum_{i=1}^m e^{v^\top v_i^-/\tau}}.$$

[Implicit feature modification] Given budget $\varepsilon \in \mathbb{R}_+^m$, and encoder $f : \mathcal{X} \to \mathbb{S}^d$, an adversary removes features from $f$ that discriminates batch $x, x^+, \{x_i^-\}_{i=1}^m$ by maximizing the point-wise InfoNCE loss, $\ell_\varepsilon(v, v^+, \{v_i^-\}_{i=1}^m) = \max_{\delta^+ \in \mathcal{B}_{\varepsilon^+}, \{\delta_i^- \in \mathcal{B}_{\varepsilon_i}\}_{i=1}^m} \ell(v, v^+ + \delta^+, \{v_i^- + \delta_i^-\}_{i=1}^m)$. Here $\mathcal{B}_\varepsilon$ denotes the $\ell_2$-ball of radius $\varepsilon$. Implicit feature modification (IFM) removes components of the current representations that are used to discriminate positive and negative pairs. In other words, the embeddings of positive and negative samples are modified to remove well represented features. So, if the encoder is currently using a simple shortcut solution, IFM removes the features used, thereby encouraging the encoder to also discriminate instances using other features. By applying perturbations in the embedding space IFM can modify high level semantic features (see Fig. 4), which is extremely challenging when applying perturbations in input space. In order to learn new features using the perturbed loss while still learning potentially complementary information using the original InfoNCE objective, we propose optimizing the the multi-task objective $\min_f \{\mathcal{L}(f) + \alpha \mathcal{L}_\varepsilon(f)\}/2$ where $\mathcal{L}_\varepsilon = \mathbb{E}\ell_\varepsilon$ is the adversarial perturbed loss, and $\mathcal{L}$ the standard InfoNCE loss. For simplicity, all experiments set the balancing parameter $\alpha = 1$ unless explicitly noted, and all take $\varepsilon^+, \varepsilon_i^-$ to be equal, and denote this single value by $\varepsilon$. Crucially, $\ell_\varepsilon$ can be computed analytically and efficiently. For any $v, v^+, \{v_i^-\}_{i=1}^m \in \mathbb{R}^d$ we have,

$$\nabla_{v_j^-} \ell = \frac{e^{v^\top v_j^- / \tau}}{e^{v^\top v^+ / \tau} + \sum_{i=1}^m e^{v^\top v_i^- / \tau}} \cdot \frac{v}{\tau} \quad \text{and} \quad \nabla_{v^+} \ell = \left( \frac{e^{v^\top v^+ / \tau}}{e^{v^\top v^+ / \tau} + \sum_{i=1}^m e^{v^\top v_i^- / \tau}} - 1 \right) \cdot \frac{v}{\tau}.$$

In particular, $\nabla_{v_j^-} \ell \propto v$ and $\nabla_{v^+} \ell \propto -v$. This expression shows that the adversary perturbs $v_j^-$ (resp. $v^+$) in the direction of the anchor $v$ (resp $-v$). Since the derivative directions are *independent* of $\{v_i^-\}_{i=1}^m$ and $v^+$, we can analytically compute optimal perturbations in $\mathcal{B}_\varepsilon$. Indeed, following the constant ascent direction shows the optimal updates are simply $v_i^- \leftarrow v_i^- + \varepsilon_i v$ and $v^+ \leftarrow v^+ - \varepsilon^+ v$. The positive (resp. negative) perturbations increase (resp. decrease) cosine similarity to the anchor $\text{sim}(v, v_i^- + \varepsilon_i v) \to 1$ as $\varepsilon_i \to \infty$ (resp. $\text{sim}(v, v^+ - \varepsilon^+ v) \to -1$ as $\varepsilon^+ \to \infty$). In Fig. 4 we visualize the newly synthesized $v_i^-, v^+$ and find meaningful interpolation of semantics. Plugging the update rules for $v^+$ and $v_i^-$ into the point-wise InfoNCE loss yields,

$$\ell_\varepsilon(v, v^+, \{v_i^-\}_{i=1}^m) = -\log \frac{e^{(v^\top v^+ - \varepsilon^+)/\tau}}{e^{(v^\top v^+ - \varepsilon^+)/\tau} + \sum_{i=1}^m e^{(v^\top v_i^- + \varepsilon_i)/\tau}}. \tag{2}$$

In other words, IFM amounts to simply perturbing the logits – reduce the positive logit by $\varepsilon^+/\tau$ and increase negative logits by $\varepsilon_i/\tau$. From this we see that $\ell_\varepsilon$ is automatically symmetrized in the positive samples: perturbing $v$ instead of $v^+$ results in the exact same objective. Eqn. 2 shows that IFM re-weights each negative sample by a factor $e^{\varepsilon_i/\tau}$ and positive samples by $e^{-\varepsilon^+/\tau}$.

### 3.1 Visualizing implicit feature modification

With implicit feature modification, newly synthesized data points do not directly correspond to any "true" input data point. However it is still possible to visualize the effects of implicit feature modification. To do this, assume access to a memory bank of input data $\mathcal{M} = \{x_i\}_i$. A newly synthesized sample $s$ can be approximately visualized by retrieving the 1-nearest neighbour using cosine similarity $\arg\min_{x \in \mathcal{M}} \text{sim}(s, f(x))$ and viewing the image $x$ as an approximation to $s$.

Fig. 4 shows results using a ResNet-50 encoder trained using MoCo-v2 on ImageNet1K using the training set as the memory bank. For positive pair $v, v^+$ increasing $\varepsilon$ causes the semantics of $v$ and $v^+$ to diverge. For $\varepsilon = 0.1$ a different car with similar pose and color is generated, for $\varepsilon = 0.2$ the pose and color then changes, and finally for $\varepsilon = 1$ the pose, color and type of vehicle changes. For negative pair $v, v^-$ the reverse occurs. For $\varepsilon = 0.1$, $v^-$ is a vehicle with similar characteristics (number of windows, color etc.), and with $\varepsilon = 0.2$, the pose of the vehicle $v^+$ aligns with $v$. Finally for $\varepsilon = 1$ the pose and color of the perturbed negative sample become aligned to the anchor $v$. In summary, implicit feature modification successfully *modifies the feature content in positive and negative samples*, thereby altering which features can be used to discriminate instances.

**Related Work.** Several works consider adversarial contrastive learning [19, 24, 26] using PGD (e.g. FGSM) attacks to alter samples in input space. Unlike our approach, PGD-based attacks require costly inner-loop optimization. Other work takes an adversarial viewpoint in input space for other self-supervised tasks e.g., rotations and jigsaws but uses an image-to-image network to simulate FGSM/PGD attacks [31], introducing comparable computation overheads. They note that low-level

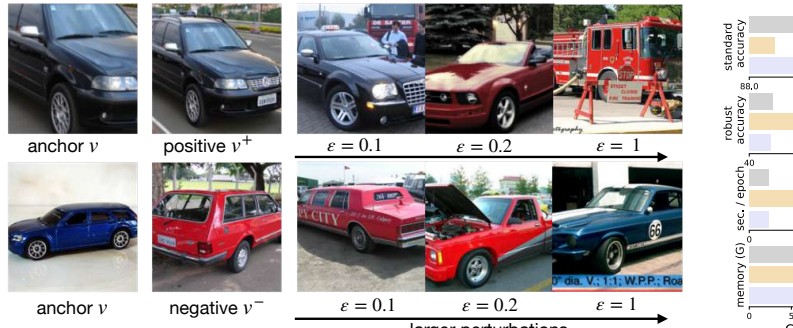
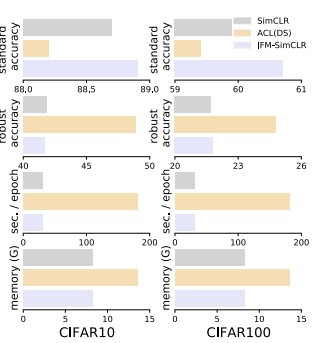

Figure 4: Visualizing implicit feature modification. **Top row:** progressively moving positive sample away from anchor. **Bottom row:** progressively moving negative sample away from anchor. In both cases, semantics such as color, orientation, and vehicle type are modified, showing the suitability of implicit feature modification for altering instance discrimination tasks.

Figure 5: Comparison between IFM and ACL(DS). Under standard linear evaluation IFM performs best. ACL is suited to adversarial evaluation.

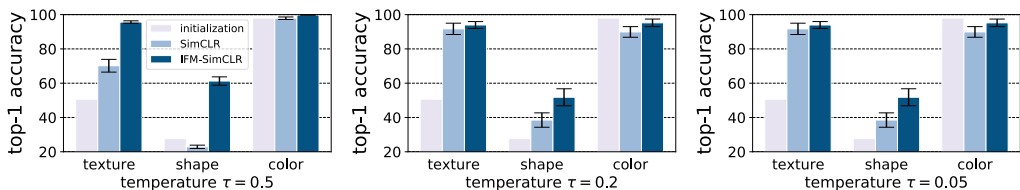

Figure 6: **Trifeature dataset.** Implicit feature modification reduces feature suppression, enhancing the representation of texture, shape and color features simultaneously. All results are average linear readout accuracy over three seeds and use a fixed value $\varepsilon = 0.1$ to illustrate robustness to $\varepsilon$.

(i.e., pixel-level) shortcuts can be avoided using their method. All of these works differ from ours by applying attacks in input space, thereby focusing on lower-level features, whereas ours aims to modify high-level features. Fig. 5 compares IFM to this family of input-space adversarial methods by comparing to a top performing method ACL(DS) [24]. We find that ACL improves robust accuracy under $\ell_\infty$-attack on input space (see [24] for protocol details), whereas IFM improves standard accuracy (full details and discussion in Appdx. C.3). Synthesizing harder negatives in latent space using Mixup [53] has also been considered [25] but does not take an adversarial perspective. Other work, AdCo [20], also takes an adversarial viewpoint in latent space. There are several differences to our approach. AdCo perturbs all negatives using the same weighted combination of all the queries, whereas IFM perturbations are query specific. In other words, IFM makes instance discrimination harder point-wise, whereas AdCo perturbation makes the InfoNCE loss larger *on average* (see Fig. 4 for visualizations of instance dependent perturbation using IFM). AdCo also treats the negatives as learnable parameters, introducing $\sim 1M$ more parameters and $\sim 7\%$ computational overhead, while IFM has no computational overhead and is implemented with only two lines of code (see Tab. 1 for empirical comparison). Finally, no previous work makes the connection between suppression of semantic features and adversarial methods in contrastive learning (see Fig. 6).

## 4 Experimental results

Implicit feature modification (IFM) can be used with any InfoNCE-based contrastive framework, and we write IFM-SimCLR, IFM-MoCo-v2 etc. to denote IFM applied within a specific framework. Code for IFM will be released publicly, and is also available in the supplementary material.

### 4.1 Does implicit feature modification help avoid feature suppression?

We study the effect IFM has on feature suppression by training ResNet-18 encoders for 200 epochs with $\tau \in \{0.05, 0.2, 0.5\}$ on the Trifeature dataset [16]. Results are averaged over three seeds, with IFM using $\varepsilon = 0.1$ for simplicity. Fig. 6 shows that IFM improves the linear readout accuracy across *all* three features for all temperature settings. The capability of IFM to enhance the representation of all features – i.e. reduce reliance on shortcut solutions – is an important contrast with tuning temperature $\tau$ or using hard negatives, which Fig. 3 shows only *trades-off* which features are learned.

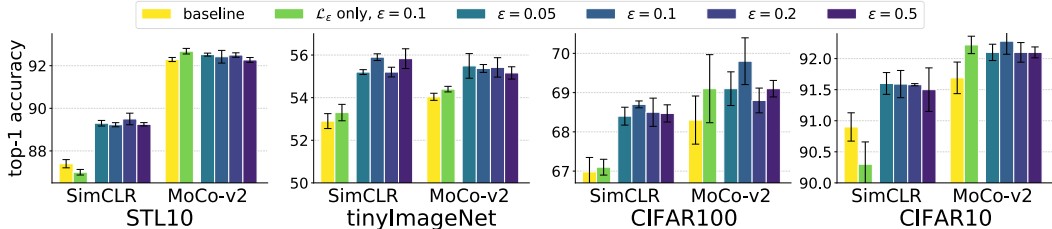

Figure 7: IFM improves linear readout performance on all datasets for all $\varepsilon \in \{0.05, 0.1, 0.2\}$ compared to baselines. Protocol uses $400$ epochs of training with ResNet-50 backbone.

## 4.2 Performance on downstream tasks

Sec. 3.1 and Sec. 4.1 demonstrate that implicit feature modification is adept at altering high-level features of an input, and combats feature suppression. This section shows that these desirable traits translate into improved performance on object classification and medical imaging tasks.

**Experimental setup for classification tasks.** Having observed the positive effect IFM has on feature suppression, we next test if this feeds through to improved performance on real tasks of interest. We benchmark using both SimCLR and MoCo-v2 [5, 7] with standard data augmentation [5]. All encoders have ResNet-50 backbones and are trained for 400 epochs (with the exception of on ImageNet100, which is trained for 200 epochs). All encoders are evaluated using the test accuracy of a linear classifier trained on the full training dataset (see Appdx. C.4 for full setup details).

**Classification tasks.** Results given in Fig. 7 and Tab. 1 find that every value of $0 < \varepsilon \le 0.2$ *improves performance across all datasets using both MoCo-v2 and Sim-CLR frameworks*. We find

| – | MoCo-v2 | AdCo [20] | IFM-MoCo-v2 | | |
|---|---|---|---|---|---|
| $\varepsilon$ | N/A | N/A | 0.05 | 0.1 | 0.2 |
| top-1 | $80.4_{\pm 0.11}$ | $78.9_{\pm 0.21}$ | $\mathbf{81.1}_{\pm 0.02}$ | $80.9_{\pm 0.25}$ | $80.7_{\pm 0.13}$ |

Table 1: Linear readout ($\%$) on ImageNet100, averaged over five seeds. IFM improves over MoCo-v2 for all settings of $\varepsilon$.

that optimizing $\mathcal{L}_\varepsilon$ ($76.0\%$ average score across all eight runs in Fig. 7) performs similarly to the standard contrastive loss ($75.9\%$ average score), and does worse than the IFM loss $(\mathcal{L} + \mathcal{L}_\varepsilon)/2$. This suggests that $\mathcal{L}$ and $\mathcal{L}_\varepsilon$ learn complementary features. Tab. 1 benchmarks IFM on ImageNet100 [44] using MoCo-v2, observing improvements of $0.9\%$. We also compare results on ImageNet100 to AdCo [20], another adversarial method for contrastive learning. We adopt the official code and use the exact same training and finetuning hyperparameters as for MoCo-v2 and IFM. For the AdCo-specific hyperparamters – negatives learning rate $lr_{\text{neg}}$ and negatives temperature $\tau_{\text{neg}}$ – we use a grid search over all combinations $lr_{\text{neg}} \in \{1, 2, 3, 4\}$ and $\tau_{\text{neg}} \in \{0.02, 0.1\}$, which includes the AdCo default ImageNet1K recommendations $lr_{\text{neg}} = 3$ and $\tau_{\text{neg}} = 0.02$ [20]. The resulting AdCo performance of $78.9\%$ is slightly below MoCo-v2. However using their respective ImageNet1K default parameters AdCo and MoCo-v2 achieve $72.4\%$ and $71.8\%$ respectively, suggesting that the discrepancy between AdCo and MoCo-v2 may in part be due to the use of improved hyperparameters tuned on MoCo-v2. Note importantly, IFM is robust to the choice of $\varepsilon$: all values $\varepsilon \in \{0.05, 0.1, 0.2\}$ were found to boost performance across all datasets and all frameworks. We emphasize that the MoCo-v2 baseline performance of $80.5\%$ on ImageNet100 is strong. Our hyperparameters, which we detail in Appdx. C.4.1, may be of interest to other works benchmarking MoCo-v2 on ImageNet100.

**Medical images.** To evaluate our method on a modality differing significantly from object-based images we consider the task of learning representations of medical images. We benchmark using the approach proposed by [42] which is a variant of MoCo-v2 that incorporates the anatomical context in the medical images. We evaluate our method on the COPDGene dataset [38], which is a multi-center observational study focused on the genetic epidemiology of Chronic obstructive pulmonary disease (COPD). See Appdx. C.5 for full background details on the COPDGene dataset, the five COPD related outcomes we use for evaluation, and our implementation. We perform regression analysis for continuous outcomes in terms of coefficient of determination (R-square), and logistic regression to predict ordinal outcomes and report the classification accuracy and the *1-off* accuracy, i.e., the probability of the predicted category is within one class of true value.

Tab. 2 reports results. For fair comparison we use same experimental configuration for the baseline approach [42] and our method. We find that IFM yields improvements on all outcome predictions.

| Method | logFEV1pp | logFEV$_1$FVC | CLE | CLE *1-off* | Para-septal | Para-septal *1-off* | mMRC | mMRC *1-off* |
|---|---|---|---|---|---|---|---|---|
| Loss | R-Square | | Accuracy (%) | | | | | |
| $\mathcal{L}$ (baseline) | $0.566_{\pm.005}$ | $0.661_{\pm.005}$ | $49.6_{\pm0.4}$ | $81.8_{\pm0.5}$ | $55.7_{\pm0.3}$ | $84.4_{\pm0.2}$ | $50.4_{\pm0.5}$ | $72.5_{\pm0.3}$ |
| $\mathcal{L}_\varepsilon, \varepsilon = 0.1$ | $0.591_{\pm.008}$ | $0.681_{\pm.008}$ | $49.4_{\pm0.4}$ | $81.9_{\pm0.3}$ | $55.6_{\pm0.3}$ | $85.1_{\pm0.2}$ | $50.3_{\pm0.8}$ | $72.7_{\pm0.4}$ |
| IFM, $\varepsilon = 0.1$ | $0.615_{\pm.005}$ | $0.691_{\pm.006}$ | $48.2_{\pm0.8}$ | $80.6_{\pm0.4}$ | $55.3_{\pm0.4}$ | $84.7_{\pm0.3}$ | $50.4_{\pm0.5}$ | $72.8_{\pm0.2}$ |
| IFM, $\varepsilon = 0.2$ | $0.595_{\pm.006}$ | $0.683_{\pm.006}$ | $48.5_{\pm0.6}$ | $80.5_{\pm0.6}$ | $55.3_{\pm0.3}$ | $85.1_{\pm0.1}$ | $49.8_{\pm0.8}$ | $72.0_{\pm0.3}$ |
| IFM, $\varepsilon = 0.5$ | $0.607_{\pm.006}$ | $0.683_{\pm.005}$ | $49.6_{\pm0.4}$ | $82.0_{\pm0.3}$ | $54.9_{\pm0.2}$ | $84.7_{\pm0.2}$ | $50.6_{\pm0.4}$ | $73.1_{\pm0.2}$ |
| IFM, $\varepsilon = 1.0$ | $0.583_{\pm.005}$ | $0.675_{\pm.006}$ | $50.0_{\pm0.5}$ | $82.9_{\pm0.4}$ | $56.3_{\pm0.6}$ | $85.7_{\pm0.2}$ | $50.3_{\pm0.6}$ | $71.9_{\pm0.3}$ |

Table 2: Linear readout performance on COPDGene dataset. The values are the average of 5-fold cross validation with standard deviations. The bold face indicates the best average performance. IFM yields improvements on all phenotype predictions.

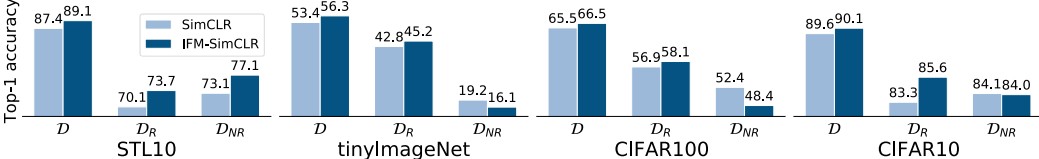

Figure 8: Label $\{\mathcal{D}, \mathcal{D}_\mathrm{R}, \mathcal{D}_\mathrm{NR}\}$ indicates which dataset was used to train the linear readout function. Improved performance of IFM on standard data $\mathcal{D}$ can be attributed to improved representation of *robust* features $\mathcal{D}_\mathrm{R}$. See Sec. 4.3 for construction of robust ($\mathcal{D}_\mathrm{R}$) and non-robust ($\mathcal{D}_\mathrm{NR}$) datasets.

The gain is largest on spirometry outcome prediction, particularly logFEV1pp with improvement of $8.7\%$ with $\varepsilon = 0.1$. We found that at least $\varepsilon = 0.5$ and $1.0$ improve performance on all tasks. However, we note that not all features yield a statistically significant improvement with IFM.

### 4.3 Further study on the impact of IFM on feature learning

This section further studies the effect implicit feature modification has on *what type* of features are extracted. Specifically, we consider the impact on learning of robust (higher-level) vs. non-robust features (pixel-level features). Our methodology, which is similar to that of Ilyas et al. [22] for deep supervised learning, involves carefully perturbing inputs to obtain non-robust features.

**Constructing non-robust features.** Given encoder $f$ we finetune a linear probe (classifier) $h$ on-top of $f$ using training data (we do not use data augmentation). Once $h$ is trained, we consider each labeled example $(x, y)$ from training data $\mathcal{D}_\mathrm{train} \in \{\text{tinyImageNet, STL10, CIFAR10, CIFAR100}\}$. A hallucinated target label $t$ is sampled uniformly at random, and we perturb $x = x_0$ until $h \circ f$ predicts $t$ using repeated FGSM attacks [12] $x_k \leftarrow x_{k-1} - \varepsilon \text{sign}(\nabla_x \ell(h \circ f(x_{k-1}), t))$. At each step we check if $\arg\max_i h \circ f(x_k)_i = t$ (we use the maximum of logits for inference) and stop iterating and set $x_\mathrm{adv} = x_k$ for the first $k$ for which the prediction is $t$. This usually takes no more than a few FGSM steps with $\varepsilon = 0.01$. We form a dataset of "robust" features by adding $(x_\mathrm{adv}, y)$ to $\mathcal{D}_R$, and a dataset of "non-robust" features by adding $(x_\mathrm{adv}, t)$ to $\mathcal{D}_{NR}$. To a human the pair $(x_\mathrm{adv}, t)$ will look mislabeled, but for the encoder $x_\mathrm{adv}$ contains features predictive of $t$. Finally, we re-finetune (i.e. re-train) linear classifier $g$ using $\mathcal{D}_R$ (resp. $\mathcal{D}_{NR}$).

Fig. 8 compares accuracy of the re-finetuned models on a test set of *standard* $\mathcal{D}_\mathrm{test}$ examples (no perturbations are applied to the test set). Note that $\mathcal{D}_R, \mathcal{D}_{NR}$ depend on the original encoder $f$. When re-finetuning $f$ we always use datasets $\mathcal{D}_R, \mathcal{D}_{NR}$ formed via FGSM attacks on $f$ itself. So there is one set $\mathcal{D}_R, \mathcal{D}_{NR}$ for SimCLR, and another set for IFM. Fig. 8 shows that IFM achieves superior generalization ($\mathcal{D}$) compared to SimCLR *by better representing robust features* ($\mathcal{D}_R$). Representation of non-robust features ($\mathcal{D}_{NR}$) is similar for IFM ($55.5\%$ average across all datasets) and SimCLR ($56.7\%$ average). IFM is juxtaposed to the supervised adversarial training of Madry et al., which *sacrifices* standard supervised performance in exchange for not using non-robust features [30, 46].

## 5 Discussion

This work studies the relation between contrastive instance discrimination and feature learning. While we focus specifically on contrastive learning, it would be of interest to also study feature learning for other empirically successful self-supervised methods [1, 6, 13, 51]. Understanding differences in feature learning biases between different methods may inform which methods are best suited for a given task, as well as point the way to further improved self-supervised techniques.

**Acknowledgments**   SJ was supported by NSF BIGDATA award IIS-1741341, NSF Convergence Accelerator Track D 2040636. SS acknowledges support from NSF-TRIPODS+X:RES (1839258). JR was partially supported by a Two Sigma fellowship. KB acknowledges support from NIH (1R01HL141813-01), NSF (1839332 Tripod+X), and a research grant from SAP SE Commonwealth Universal Research Enhancement (CURE) program awards research grants from the Pennsylvania Department of Health. Finally, we warmly thank Katherine Hermann and Andrew Lampinen for making the Trifeature dataset available for our use.

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
