# A   Proofs for Section 2

In this section we give proofs for all the results in Sec. 2, which explores the phenomenon of feature suppression in contrastive learning using the InfoNCE loss. We invite the reader to consult Sec. 2.1 for details on any notation, terminology, or formulation details we use.

Recall, for a measure $\nu$ on a space $\mathcal{U}$ and a measurable map $h : \mathcal{U} \to \mathcal{V}$ let $h\#\nu$ denote the *pushforward* $h\#\nu(V) = \nu(h^{-1}(V))$ of a measure $\nu$ on a space $\mathcal{U}$ for a measurable map $h : \mathcal{U} \to \mathcal{V}$ and measurable $V \subseteq \mathcal{V}$, where $h^{-1}(V)$ denotes the preimage. We now recall the definition of feature suppression and distinction.

**Definition 1.** *Consider an encoder $f : \mathcal{X} \to \mathbb{S}^{d-1}$ and features $j \subseteq [n]$. For each $z^j \in \mathcal{Z}^S$, let $\mu(\cdot|z^S) = (f \circ g)\#\lambda(\cdot|z^j)$ be the pushforward measure on $\mathbb{S}^{d-1}$ by $f \circ g$ of the conditional $\lambda(\cdot|z^S)$.*

    *1. $f$ suppresses $S$ if for any pair $z^S, \bar{z}^S \in \mathcal{Z}^S$, we have $\mu(\cdot|z^S) = \mu(\cdot|\bar{z}^S)$.*

    *2. $f$ distinguishes $S$ if for any pair of distinct $z^S, \bar{z}^S \in \mathcal{Z}^S$, measures $\mu(\cdot|z^S), \mu(\cdot|\bar{z}^S)$ have disjoint support.*

Suppression of features $S$ is thereby captured by the characteristic of distributing points in the same way on the sphere independently of what value $z^S$ takes. Feature distinction, meanwhile, is characterized by being able to partition the sphere into different pieces, each corresponding to a different value of $z^S$. Other (perhaps weaker) notions of feature distinction may be useful in other contexts. However here our goal is to establish that it is possible for InfoNCE optimal encoders both to suppress features in the sense of Def. 1, but also to separate concepts out in a desirable manner. For this purpose we found this strong notion of distinguishing to suffice.

Before stating and proving the result, recall the limiting InfoNCE loss that we analyze,

$$\mathcal{L} = \lim_{m \to \infty} \left\{ \mathcal{L}_m(f) - \log m - \tfrac{2}{\tau} \right\} = \tfrac{1}{2\tau}\mathbb{E}_{x,x^+}\|f(x) - f(x^+)\|^2 + \mathbb{E}_{x^+}\log\left[\mathbb{E}_{x^-}e^{f(x^+)^\top f(x^-)/\tau}\right].$$

We subtract $\log m$ to ensure the limit is finite, and use $x^-$ to denote a random sample with the same distribution as $x_i^-$. Following [49] we denote the first term by $\mathcal{L}_{\text{align}}$ and the second term by the "uniformity loss" $\mathcal{L}_{\text{unif}}$, so $\mathcal{L} = \mathcal{L}_{\text{align}} + \mathcal{L}_{\text{unif}}$.

**Proposition 1.** *Suppose that $p_j$ is uniform on $\mathcal{Z}^j = \mathbb{S}^{d-1}$. For any feature $j \in [n]$ there exists an encoder $f_{supp}$ that suppresses feature $j$ and encoder $f_{disc}$ that discriminates $j$ but both attain $\min_{f: \text{measurable}} \mathcal{L}(f)$.*

*Proof.* The existence of the encoders $f_{\text{supp}}$ and $f_{\text{disc}}$ is demonstrated by constructing explicit examples. Before defining $f_{\text{supp}}$ and $f_{\text{disc}}$ themselves, we begin by constructing a family $\{f^k\}_{k \in [n]}$ of optimal encoders.

Since $g$ is injective, we know there exists a left inverse $h : \mathcal{X} \to \mathcal{Z}$ such that $h \circ g(z) = z$ for all $z \in \mathcal{Z}$. For any $k \in [n]$ let $\Pi^k : \mathcal{Z} \to \mathbb{S}^{d-1}$ denote the projection $\Pi^k(z) = z^k$. Since $p_k$ is uniform on the sphere $\mathbb{S}^{d-1}$, we know that $\Pi^k \circ h \circ g(z) = z^j$ is uniformly distributed on $\mathbb{S}^{d-1}$. Next we partition the space $\mathcal{X}$. Since we assume that for all $a \neq a'$ and $z \neq z'$ that $a(z) \neq a'(z')$, the family $\{\mathcal{X}_z\}_{z \in \mathcal{Z}}$ where $\mathcal{X}_z = \{a \circ g(z) : z \in \mathcal{Z}\}$ is guaranteed to be a partition (and in particular, disjoint). We may therefore define an encoder $f_k : \mathcal{X} \to \mathbb{S}^{d-1}$ to be equal to $f_k(x) = \Pi^k \circ h \circ g(z) = z^k$ for all $x \in \mathcal{X}_z$.

First we check that this $f_k$ is optimal. Since for any $z$, and any $a \sim \mathcal{A}$, by definition we have $a \circ g(z) \in \mathcal{X}_z$, we have that $f_k(x) = f_k(a(x))$ almost surely, so $\mathcal{L}_{\text{align}}(f_k) = 0$ is minimized. To show $f_k$ minimizes $\mathcal{L}_{\text{unif}}$ note that the uniformity loss can be re-written as

$$\mathcal{L}_{\text{unif}}(f_k) = \int_a \int_z \log \int_{a^-} \int_{z^-} e^{f_k \circ a(g(z))^\top f_k \circ a^-(g(z^-))/\tau} \lambda(\mathrm{d}z)\lambda(\mathrm{d}z^-)\mathcal{A}(\mathrm{d}a)\mathcal{A}(\mathrm{d}a^-)$$

$$= \int_z \log \int_{z^-} e^{f_k \circ g(z)^\top f_k \circ g(z^-)/\tau} \lambda(\mathrm{d}z)\lambda(\mathrm{d}z^-)$$

$$= \int_{\mathbb{S}^{d-1}} \log \int_{\mathbb{S}^{d-1}} e^{u^\top v/\tau} \mu(\mathrm{d}u)\mu(\mathrm{d}v)$$

where $\mu = f_k \circ g\#\lambda$ is the pushforward measure on $\mathbb{S}^{d-1}$, and the second equality follows from the fact that $\mathcal{L}_{\text{align}}(f_k) = 0$. Theorem 1 of Wang and Isola [49] establishes that the operator,

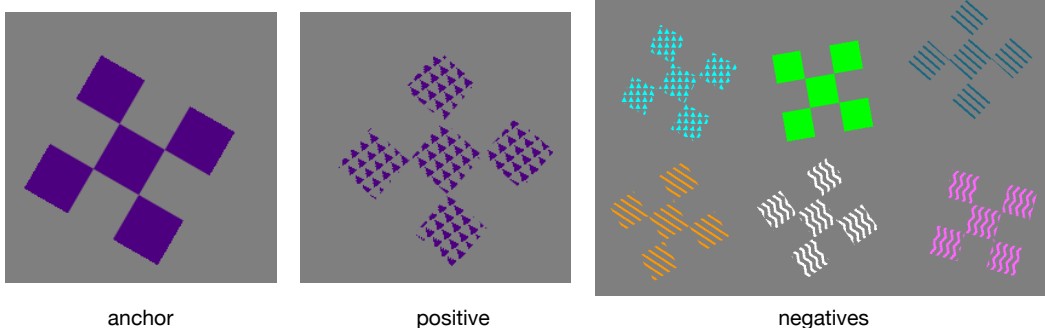

anchor         positive         negatives

Figure 9: Visual illustration of Prop. 2.3 using Trifeature samples [16]. In this example the shape feature is kept fixed across all positives and negatives ( $S = \{\text{shape}\}$ ), with color and texture to varying. As a consequence, the positive pair cannot be discriminated from negatives using the shape feature. The encoder must learn color features in order to identify this positive pair. In other words, if a given set of features (e.g. $S = \{\text{shape}\}$) are constant across positive and negative pairs, then instance discrimination task demands the use of features in the compliment (e.g. $\{\text{color,texture}\}$).

$$\mu \mapsto \int_{\mathbb{S}^{d-1}} \log \int_{\mathbb{S}^{d-1}} e^{u^\top v/\tau} \mu(\mathrm{d}u)\mu(\mathrm{d}v)$$

is minimized over the space of Borel measures on $\mathbb{S}^{d-1}$ if and only if $\mu = \sigma_d$, the uniform distribution on $\mathbb{S}^{d-1}$, as long as such an $f$ exists. However, since by construction $f_k(x) = \Pi^k \circ h \circ g(z) = z^k$ is uniformly distributed on $\mathbb{S}^{d-1}$, we know that $(f_k \circ g)\#\lambda = \sigma_d$, and hence that $f_k$ minimizes $\mathcal{L}_{\text{align}}$ and $\mathcal{L}_{\text{unif}}$ and hence also the sum $\mathcal{L} = \mathcal{L}_{\text{align}} + \mathcal{L}_{\text{unif}}$.

Recall that we seek encoder $f_{\text{supp}}$ that suppress feature $j$, and $f_{\text{disc}}$ that distinguishes feature $j$. We have a family $\{f^k\}_{k \in [n]}$ that are optimal, and select the two encoders we seen from this collection. First, for $f_{\text{supp}}$ define $f_{\text{supp}} = f^k$ for any $k \neq j$. Then by construction $f_{\text{supp}}(x) = z^k$ (where $x \in \mathcal{X}_z$) depends only on $z^k$, which is independent of $z^j$. Due to independence, we therefore know that for any pair $z^j, \bar{z}^j \in \mathcal{Z}^j$, we have $\mu(\cdot|z^j) = \mu(\cdot|\bar{z}^j)$, i.e., that $f_{\text{supp}}$ is optimal but suppresses feature $j$. Similarly, simply define $f_{\text{disc}} = f^j$. So $f_{\text{disc}}(x) = z^j$ where $x \in \mathcal{X}_z$, and for any $z^j, \bar{z}^j \in \mathcal{Z}^j$ with $z^j \neq \bar{z}^j$ the pushforwards $\mu(\cdot|z^j), \mu(\cdot|\bar{z}^j)$ are the Dirac measures $\delta_{z^j}, \delta_{\bar{z}^j}$, which are disjoint. $\square$

Next we present a result showing that, under suitable conditions that guarantee that minimizers exists, any $f$ optimizing the InfoNCE loss is guaranteed to suppress features $S$ if all batches $x_1^+, x_2^+, \{x_i^-\}_{i=1}^N$ are have the same features $S$ (but that the value $z_S$ taken is allowed to vary). This result captures the natural intuition that if a feature cannot be used to discriminate instances, then it will not be learned by the InfoNCE loss. Before reading the proposition, we encourage the reader to see Fig. 9 for an intuitive visual illustration of the idea underlying Prop. 2.3 using Trifeature samples [16].

However, this result also points to a way to manage which features are learned by an encoder, since if $f$ is guaranteed *not* to learn features $S$, then necessarily $f$ must use other features to solve the instance discrimination task. This insight lays the foundation for the implicit feature modification technique, which perturbs the embedding $v = f(x)$ to remove information that $f$ uses to discriminate instances – and then asks for instance discrimination using *both* to original embedding, and the modified one – with the idea that this encourages $f$ to learn new features that it previously suppressed.

**Proposition 2.** *For a set $S \subseteq [n]$ of features let*

$$\mathcal{L}_S(f) = \mathcal{L}_{align}(f) + \mathbb{E}_{x^+}\big[ - \log \mathbb{E}_{x^-}[e^{f(x^+)^\top f(x^-)}|z^S = z^{S-}]\big]$$

*denote the (limiting) InfoNCE conditioned on $x^+, x^-$ having the same features $S$. Suppose that $p_j$ is uniform on $\mathcal{Z}^j = \mathbb{S}^{d-1}$ for all $j \in [n]$. Then the infimum $\inf \mathcal{L}_S$ is attained, and every $f \in \min_{f'} \mathcal{L}_S(f')$ suppresses features $S$ almost surely.*

*Proof.* By Prop 2.3, we know that for each $z^S$ there is a measurable $f$ such that $\mathcal{L}_{\text{align}}(f) = 0$ and $f$ achieves perfect uniformity $(f \circ g)\#\lambda(\cdot|z^S) = \sigma_d$ conditioned on $z^S$. So consider such an $f$. Since $\mathcal{L}_{\text{align}}(f) = 0$ we may write,

$$\mathcal{L}_S(f) = \mathbb{E}_{x^+}\big[-\log \mathbb{E}_{x^-}[e^{f(x^+)^\top f(x^-)}|z^S = z^{S-}]\big]$$
$$= \mathbb{E}_{z^S}\mathbb{E}_{z^{S-}}\big[-\log \mathbb{E}_{z^-}[e^{f\circ g(z)^\top f\circ g(z^-)}|z^S = z^{S-}]\big]$$
$$= \mathbb{E}_{z^S}\mathcal{L}(f; z^S).$$

Where we have introduced the conditional loss function

$$\mathcal{L}(f; z^S) = \mathbb{E}_{z^{S-}}\big[-\log \mathbb{E}_{z^-}[e^{f\circ g(z)^\top f\circ g(z^-)}|z^S = z^{S-}]\big]$$

We shall show that any minimizer $f$ of $\mathcal{L}_S$ is such that $f$ minimizes $\mathcal{L}(f; z^S)$ for all values of $z^S$. To show this notice that $\min_f \mathcal{L}_S(f) = \min_f \mathbb{E}_{z^S}\mathcal{L}(f; z^S) \geq \mathbb{E}_{z^S}\min_f \mathcal{L}(f; z^S)$ and if there is an $f$ such that $f$ minimizes $\mathcal{L}(f; z^S)$ for each $z^S$ then the inequality is tight. So we make it our goal to show that there is an $f$ such that $f$ minimizes $\mathcal{L}(f; z^S)$ for each $z^S$.

For fixed $z^S$, by assumption there is an $f_{z^S}$ such that $(f_{z^S} \circ g)\#\lambda(\cdot|z^S) = \sigma_d$. That is, $f_{z^S}$ achieves perfect uniformity given $z^S$. Theorem 1 of Wang and Isola [49] implies that $f_{z^S}$ must minimize $\mathcal{L}(f; z^S)$. Given $\{f_{z^S}\}_{z^S}$ we construct an $f : \mathcal{X} \to \mathbb{S}_\tau^{d-1}$ that minimizes $\mathcal{L}(f; z^S)$ *for all $z^S$*. By injectivity of $g$ we may partition $\mathcal{X}$ into pieces $\bigcup_{z^S \in \mathcal{Z}^S} \mathcal{X}_{z^S}$ where $\mathcal{X}_{z^S} = \{x : x = g((z^S, z^{S^c}))$ for some $z^{S^c} \in \mathcal{Z}^{S^c}\}$. So we may simply define $f$ on domain $\mathcal{X}$ as follows: $f(x) = f_{z^S}(x)$ if $x \in \mathcal{X}_{z^S}$.

This construction allows us to conclude that the minimum of $\mathcal{L}_S$ is attained, and any minimizer $f$ of $\mathcal{L}_S$ also minimizes $\mathcal{L}(f; z^S)$ for each $z^S$. By Theorem 1 of Wang and Isola [49] any such $f$ is such that $(f_{z^S} \circ g)\#\lambda(\cdot|z^S) = \sigma_d$ for all $z^S$, which immediately implies that $f$ suppresses features $S$. $\quad\square$

# B  Computation of implicit feature modification updates

This section gives detailed derivations of two simple but key facts used in the development of IFM. The first result derives an analytic expression for the gradient of the InfoNCE loss with respect to positive sample in latent space, and the second result computes the gradient with respect to an arbitrary negative sample. The analysis is very simple, only requiring the use of elementary tools from calculus. Despite its simplicity, this result is very important, and forms the core of our approach. It is thanks to the analytic expressions for the gradients of the InfoNCE loss that we are able to implement our adversarial method *without introducing any memory or run-time overheads*. This is a key distinction from previous adversarial methods for contrastive learning, which introduce significant overheads (see Fig. 5).

Recall the statement of the lemma.

For any $v, v^+, \{v_i^-\}_{i=1}^m \in \mathbb{R}^d$ we have,

$$\nabla_{v_j^-}\ell = \frac{e^{v^\top v_j^-}}{e^{v^\top v^+/\tau} + \sum_{i=1}^m e^{v^\top v_i^-/\tau}} \cdot \frac{v}{\tau} \quad \text{and} \quad \nabla_{v^+}\ell = \left(\frac{e^{v^\top v^+/\tau}}{e^{v^\top v^+/\tau} + \sum_{i=1}^m e^{v^\top v_i^-/\tau}} - 1\right) \cdot \frac{v}{\tau}.$$

In particular, $\nabla_{v_j^-}\ell \propto v$ and $\nabla_{v^+}\ell \propto -v$.

*Proof.* Both results follow from direct computation. First we compute $\nabla_{v_j^-}\ell(v, v^+, \{v_i^-\}_{i=1}^m)$. Indeed, for any $j \in \{1, 2, \ldots, m\}$ we have,

$$\nabla_{v_j^-}\left\{-\log \frac{e^{v^\top v^+/\tau}}{e^{v^\top v^+/\tau} + \sum_{i=1}^m e^{v^\top v_i^-/\tau}}\right\} = \nabla_{v_j^-}\log\left\{e^{v^\top v^+/\tau} + \sum_{i=1}^m e^{v^\top v_i^-/\tau}\right\}$$

$$= \frac{\nabla_{v_j^-}\left\{e^{v^\top v^+} + \sum_{i=1}^m e^{v^\top v_i^-/\tau}\right\}}{e^{v^\top v^+/\tau} + \sum_{i=1}^m e^{v^\top v_i^-/\tau}}$$

$$= \frac{e^{v^\top v_j^-/\tau} \cdot v/\tau}{e^{v^\top v^+/\tau} + \sum_{i=1}^m e^{v^\top v_i^-/\tau}}$$

the quantity $\frac{e^{v^\top v_j^-/\tau}}{e^{v^\top v^+/\tau}+\sum_{i=1}^m e^{v^\top v_i^-/\tau}} > 0$ is a strictly positive scalar, allowing us to conclude the derivative $\nabla_{v_j^-}\ell$ is proportional to $v$. We also compute $\nabla_{v^+}\ell(v, v^+, \{v_i^-\}_{i=1}^m)$ in a similar fashion,

$$\nabla_{v^+}\left\{-\log \frac{e^{v^\top v^+/\tau}}{e^{v^\top v^+/\tau}+\sum_{i=1}^m e^{v^\top v_i^-/\tau}}\right\} = \nabla_{v^+}\left\{-\log e^{v^\top v^+/\tau}\right\} + \nabla_{v^+}\log\left\{e^{v^\top v^+/\tau}+\sum_{i=1}^m e^{v^\top v_i^-/\tau}\right\}$$

$$= -\frac{v}{\tau} + \frac{\nabla_{v^+}\left\{e^{v^\top v^+/\tau}+\sum_{i=1}^m e^{v^\top v_i^-/\tau}\right\}}{e^{v^\top v^+/\tau}+\sum_{i=1}^m e^{v^\top v_i^-/\tau}}$$

$$= -\frac{v}{\tau} + \frac{e^{v^\top v^+/\tau}\cdot v/\tau}{e^{v^\top v^+/\tau}+\sum_{i=1}^m e^{v^\top v_i^-/\tau}}$$

$$= \left(\frac{e^{v^\top v^+/\tau}}{e^{v^\top v^+/\tau}+\sum_{i=1}^m e^{v^\top v_i^-/\tau}} - 1\right)\cdot\frac{v}{\tau}.$$

Since $0 < \frac{e^{v^\top v^+/\tau}}{e^{v^\top v^+/\tau}+\sum_{i=1}^m e^{v^\top v_i^-/\tau}} < 1$ we conclude in this case that the derivative $\nabla_{v^+}\ell$ points in the direction $-v$. $\qquad\square$

## B.1 Alternative formulations of implicit feature modification

This section contemplates two simple modifications to the IFM method with the aim of confirming that these modifications do not yield superior performance to the default proposed method. The two alternate methods focus around the following observation: IFM perturbs embeddings of unit length, and returns a modified version that will no longer be of unit length in general. We consider two alternative variations of IFM that yield normalized embeddings. The first is the most simple solution possible: simply re-normalize perturbed embeddings to have unit length. The second is slightly more involved, and involves instead applying perturbations *before* normalizing the embeddings. Perturbing unnormalized embeddings, then normalizing, guarantees the final embeddings have unit length. The key property we observed in the original formulation was the existence of an analytic, easily computable closed form expressions for the derivatives. This property enables efficient computation of newly synthesized "adversarial" samples in latent space. Here we derive corresponding formulae for the pre-normalization attack.

For clarity, we introduce the slightly modified setting in full detail. We are given positive pair $x, x^+$ and a batch of negative samples $\{x_i^-\}_{i=1}^m$ and denote their encodings via $f$ as $v = f(x), v^+ = f(x^+)$, and $v_i^- = f(x_i^-)$ for $i = 1, \ldots m$ where we *do not* assume that $f$ returns normalized vectors. That is, $f$ is allowed to map to anywhere in the ambient latent space $\mathbb{R}^d$. The re-parameterized point-wise contrastive loss for this batch of samples is

$$\ell(v, v^+, \{v_i^-\}_{i=1}^m) = -\log \frac{e^{\mathrm{sim}(v,v^+)/\tau}}{e^{\mathrm{sim}(v,v^+)/\tau}+\sum_{i=1}^m e^{\mathrm{sim}(v,v_i^-)/\tau}},$$

where $\mathrm{sim}(u, v) = u \cdot v/\|u\|\|v\|$ denotes the cosine similarity measure. As before we wish to perturb $v^+$ and negative encodings $v_j^-$ to increase the loss, thereby making the negatives harder. Specifically we wish to solve $\max_{\delta^+\in\mathcal{B}_{\varepsilon^+},\{\delta_i^-\in\mathcal{B}_{\varepsilon_i}\}_{i=1}^m} \ell(v, v^+ + \delta^+, \{v_i^- + \delta_i^-\}_{i=1}^m)$. The following lemma provides the corresponding gradient directions.

For any $v, v^+, \{v_i^-\}_{i=1}^m \in \mathbb{R}^d$ we have

$$\nabla_{v_j^-}\ell \propto \frac{v}{\|v\|} - \mathrm{sim}(v_j^-, v)\frac{v_j^-}{\|v_j^-\|} \quad \text{and} \quad \nabla_{v^+}\ell \propto \frac{v}{\|v\|} - \mathrm{sim}(v^+, v)\frac{v^+}{\|v^+\|}.$$

To prove this lemma we rely on the following well-known closed form expression for the derivative of the cosine similarity, whose proof we omit.

$$\nabla_v\mathrm{sim}(v, u) = \frac{u}{\|v\|\|u\|} - \mathrm{sim}(v, u)\frac{v}{\|v\|^2}.$$

*Proof of Lemma B.1.* We compute,

$$\nabla_{v_j^-} \ell = \nabla_{v_j^-} \log \left( e^{\mathrm{sim}(v,v^+)} + \sum_{i=1}^m e^{\mathrm{sim}(v,v_i^-)} \right)$$

$$= \frac{e^{\mathrm{sim}(v,v_j^-)}}{e^{\mathrm{sim}(v,v^+)} + \sum_{i=1}^m e^{\mathrm{sim}(v,v_i^-)}} \cdot \nabla_{v_j^-} \mathrm{sim}(v, v_j^-)$$

Using the formula for the derivative of the cosine similarity, we arrive at a closed form formula,

$$\nabla_{v_j^-} \ell = \frac{e^{\mathrm{sim}(v,v_j^-)}}{e^{\mathrm{sim}(v,v^+)} + \sum_{i=1}^m e^{\mathrm{sim}(v,v_i^-)}} \cdot \left( \frac{v}{\|v_j^-\|\|v\|} - \mathrm{sim}(v_j^-, v) \frac{v_j^-}{\|v_j^-\|^2} \right).$$

$$\propto \frac{v}{\|v\|} - \mathrm{sim}(v_j^-, v) \frac{v_j^-}{\|v_j^-\|}$$

Similar computations yield

$$\nabla_{v^+} \ell = -\nabla_{v^+} \log \frac{e^{\mathrm{sim}(v,v^+)}}{e^{\mathrm{sim}(v,v^+)} + \sum_{i=1}^m e^{\mathrm{sim}(v,v_j^-)}}$$

$$= \nabla_{v^+} \left( -\mathrm{sim}(v, v^+) + \log \left( e^{\mathrm{sim}(v,v^+)} + \sum_{i=1}^m e^{\mathrm{sim}(v,v_i^-)} \right) \right)$$

$$= \left( \frac{e^{\mathrm{sim}(v,v^+)}}{e^{\mathrm{sim}(v,v^+)} + \sum_{i=1}^m e^{\mathrm{sim}(v,v_i^-)}} - 1 \right) \cdot \nabla_{v^+} \mathrm{sim}(v, v^+)$$

$$= \left( \frac{e^{\mathrm{sim}(v,v^+)}}{e^{\mathrm{sim}(v,v^+)} + \sum_{i=1}^m e^{\mathrm{sim}(v,v_i^-)}} - 1 \right) \cdot \left( \frac{v}{\|v^+\|\|v\|} - \mathrm{sim}(v^+, v) \frac{v^+}{\|v^+\|^2} \right)$$

$$\propto \frac{v}{\|v\|} - \mathrm{sim}(v^+, v) \frac{v^+}{\|v^+\|}$$

$\square$

Lemma B.1 provides precisely the efficiently computable formulae for the derivatives we seek. One important difference between this pre-normalization case and the original setting is that the direction vector depends on $v_j^-$ and $v^+$ respectively. In the original (unnormalized) setting the derivatives depend only on $v$, which allowed the immediate and exact discovery of the worst case perturbations in an $\varepsilon$-ball. Due to these additional dependencies in the pre-normalized case the optimization is more complex, and must be approximated iteratively. Although only approximate, it is still computationally cheap since we have simple analytic expressions for gradients.

It is possible give an interpretation to the pre-normalization derivatives $\nabla_{v_j^-} \ell$ by considering the $\ell_2$ norm,

$$\|\nabla_{v_j^-}\|_2 = \sqrt{ \left( \frac{v}{\|v\|} - \frac{v^\top v_j^-}{\|v\|\|v_j^-\|} \frac{v_j^-}{\|v_j^-\|} \right) \cdot \left( \frac{v}{\|v\|} - \frac{v^\top v_j^-}{\|v\|\|v_j^-\|} \frac{v_j^-}{\|v_j^-\|} \right) }$$

$$= \sqrt{ 1 + \mathrm{sim}(v, v_i^-)^2 - 2\mathrm{sim}(v, v_i^-)^2 }$$

$$= \sqrt{ 1 - \mathrm{sim}(v, v_i^-)^2 }$$

So, samples $v_i^-$ with higher cosine similarity with anchor $v$ receive smaller updates. Similar calculations for $v^+$ show that higher cosine similarity with anchor $v$ leads to larger updates. In other words, the pre-normalization version of the method automatically adopts an adaptive step size based on sample importance.

| Dataset | MoCo-v2 | IFM-MoCo-v2 | | |
|---|---|---|---|---|
| – | – | default | + norm | + pre-norm |
| STL10 | 92.4% | 92.9% | 92.9% | **93.0%** |
| CIFAR10 | 91.8% | **92.4%** | 92.2% | 92.0% |
| CIFAR100 | 69.0% | **70.3%** | 70.1% | 70.2% |

Table 3: Linear readout performance of alternative latent space adversarial methods. We report the best performance over runs for $\varepsilon \in \{0.05, 0.1, 0.2, 0.5\}$. We find that the two modifications to IFM we considered do not improve performance compared to the default version of IFM.

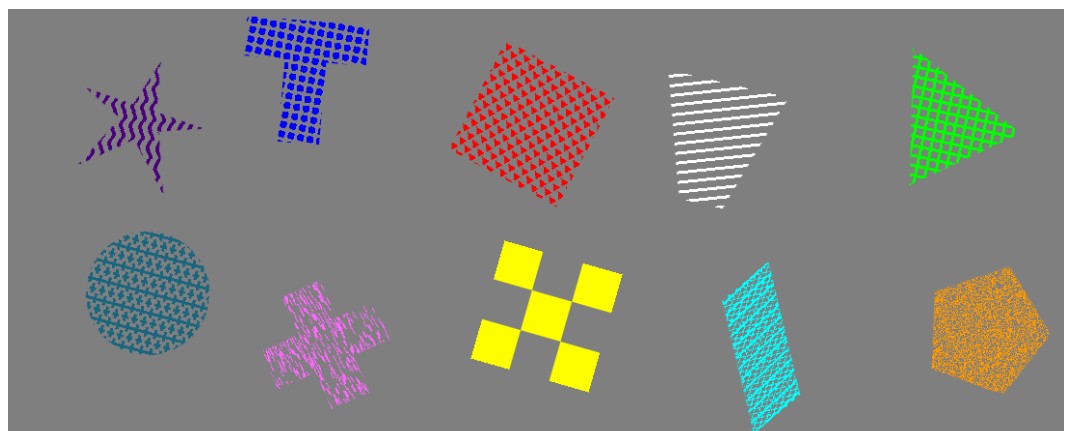

Figure 10: Sample images from the Trifeature dataset [16]. There are three features: shape, color, and texture. Each feature has 10 different possible values. We show exactly one example of each feature.

### B.1.1 Experimental results using alternative formulations

In this section we test the two alternative implementations to confirm that these simple alternatives do not obtain superior performance to IFM. We consider only object-based images, so it remains possible that other modalities may benefit from alternate formulations. First note that $f$ encodes all points to the boundary of the same hypersphere, while perturbing $v_i^- \leftarrow v_i^- + \varepsilon_i v$ and $v^+ \leftarrow v^+ - \varepsilon_+ v$ moves adversarial samples off this hypersphere. We therefore consider normalizing all points again *after* perturbing (+ *norm*). The second method considers applying attacks *before* normalization (+ *pre-norm*), whose gradients were computed in the Lem. B.1. It is still possible to compute analytic gradient expressions in this setting; we refer the reader to Appendix B.1 for full details and derivations. Results reported in Tab. 3, suggest that all versions improve over MoCov2, and both alternatives perform comparably to the default implementation based on Eqn. 2.

## C  Supplementary experimental results and details

### C.1  Hardware and setup

Experiments were run on two internal servers. The first consists of 8 NVIDIA GeForce RTX 2080 Ti GPUs (11GB). The second consists of 8 NVIDIA Tesla V100 GPUs (32GB). All experiments use the PyTorch deep learning framework [34]. Specific references to pre-existing code bases used are given in the relevant sections below.

### C.2  Feature suppression experiments

This section gives experimental details for all experiments in Sec. 2 in the main manuscript, the section studying the relation between feature suppression and instance discrimination.

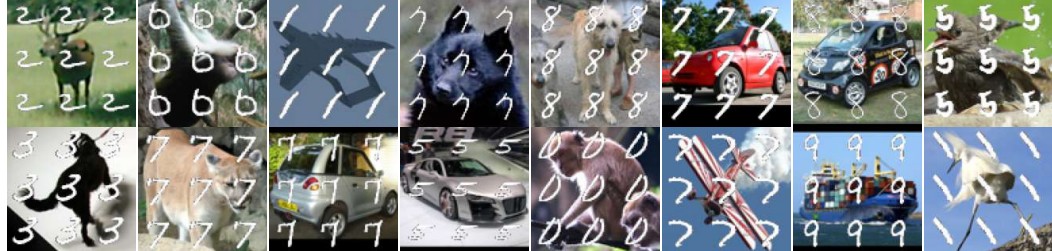

Figure 11: Sample images from the STL-digits dataset. There are two features: object class, and MNIST digit. Both features have 10 different possible values.

### C.2.1 Datasets

**Trifeature [16]**  Introduced by Hermann and Lampinen, each image is $128 \times 128$ and has three features: color, shape, and texture each taking 10 values. For each (color, shape, texture) triplet (1000 in total) Trifeature contains 100 examples, forming a dataset of 100K examples in total. Train/val sets are obtained by a random 90/10 split. See Fig. 10, Appdx. C for sample images.

**STL10-digits dataset**  We artificially combine MNIST digits and STL10 object to produce data with two controllable semantic features. We split the STL10 image into a $3 \times 3$ grid, placing a copy of the MNIST digit in the center of each sector. This is done by masking all MNIST pixels with intensity lower than 100, and updating non-masked pixels in the STL10 image with the corresponding MNIST pixel value.

### C.2.2 Experimental protocols

**Training**  We train ResNet-18 encoders using SimCLR with batch size 512. We use standard data SimCLR augmentations [5], but remove grayscaling and color jittering when training on Trifeature in order to avoid corrupting color features. We use Adam optimizer, learning rate $1 \times 10^{-3}$ and weight decay $1 \times 10^{-6}$. Unless stated otherwise, the temperature $\tau$ is set to 0.5.

**Linear evaluation**  For fast linear evaluation we first extract features from the trained encoder (applying the same augmentations to inputs as used during pre-training) then use the `LogisticRegression` function in scikit-learn [36] to train a linear classifier. We use the Limited-memory Broyden–Fletcher–Goldfarb–Shanno algorithm with a maximum iteration of 500 for training.

### C.2.3 Details on results

**Correlations Fig. 2**  For the Trifeature heatmap 33 encoders are used to compute correlations. The encoders are precisely encoders used to plot Fig. 3. Similarly, the 7 encoders used to generate the STL-digits heatmap are precisely the encoders whose training is shown in Fig. 13. When computing the InfoNCE loss for Fig. 2, for fair comparison all losses are computed using temperature normalization value $\tau = 0.5$. This is independent of training, and is necessary only in evaluation to ensure loss values are comparable across different temperatures.

Fig. 13 displays results for varying instance discrimination difficult on the STL-digits dataset. These results are complementing the Trifeature results in Fig. 3 in Sec. 2 in the main manuscript. For STL-digits we report only a single training run per hyperparameter setting since performance is much more stable on STL-digits compared to Trifeature (see Fig. 12). See Sec. 2 for discussion of STL-digits results, which are qualitatively the same as on Trifeature. Finally, Fig. 14 shows the effect of IFM on encoders trained on STL-digits. As with Trifeature, we find that IFM improves the performance on suppressed features (STL10), but only slightly. Unlike hard instance discrimination methods, IFM does not harm MNIST performance in the process.

### C.3 Comparing IFM and ACL(DS)

We give details for Fig 5. Similarly to concurrent work [19, 26], ACL [24] directly performs PGD attacks in input space. We compare to the top performing version ACL(DS) – which uses a duel stream

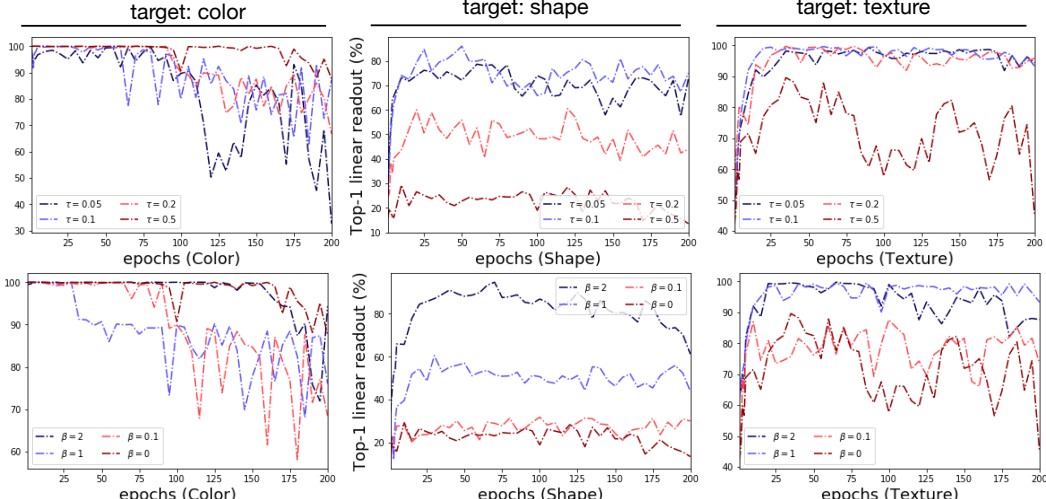

Figure 12: Single run experiments showing training dynamics of Trifeature contrastive training. Linear readout performance on color prediction is particularly noisy.

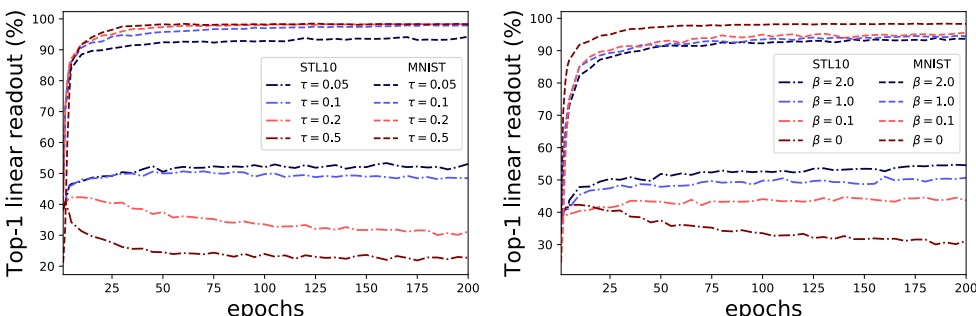

Figure 13: STL-digits dataset. **Left:** performance on STL10 and MNIST linear readout for different temperature $\tau$ values. **Right:** performance on STL10 and MNIST linear readout for different hardness concentration $\beta$ values [40]. In both cases harder instance discrimination (smaller $\tau$, bigger $\beta$) improves STL10 performance at the expense of MNIST. When instance discrimination is too easy (big $\tau$, small $\beta$) STL10 features are *suppressed*: achieving worse linear readout after training than at initialization.

structure and combines standard and adversarial loss terms. We use the official ACL implementation[1] and for fair comparison run IFM by changing only the loss function. All hyperparameters are kept the same for both runs, and follow the ACL recommendations.

**Training**  We use the SimCLR framework with a ResNet-18 backbone and train for 1000 epochs. We use a base learning rate of 5 with cosine annealing scheduling and batch size 512. LARS optimizer is used. For ACL(DS), we run the PGD for 5 steps in the pre-training stage following the practice of [24].

**Linear evaluation**  We use two schemes to evaluate the quality of learnt representation: standard accuracy and robust accuracy. Robust accuracy reports the accuracy in the setting where an adversary is allowed to apply an $\ell_\infty$ attack to each input. For standard accuracy, we only finetune the last layer and test on clean images following the practice of MoCo-v2 [7]. The initial learning rate is set as 0.1 and we tune for 100 epochs for CIFAR10, 25 epochs for CIFAR100 respectively. An SGD optimizer is used to finetune the model. We use a step scheduler that decreases the learning rate by a factor of 10 after epochs: $40, 60$ for CIFAR10; $15, 20$ for CIFAR100 respectively. For robust accuracy, we finetune the model using the loss in TRADE [52], and evaluate classification accuracy on adversarially perturbed testing images. We use the same hyperparameters as ACL [24]

---

[1] https://github.com/VITA-Group/Adversarial-Contrastive-Learning

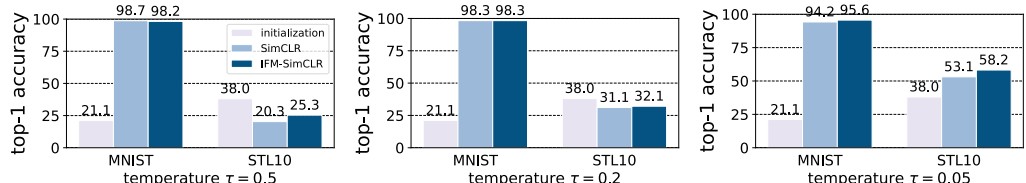

Figure 14: STL-digits dataset. Implicit feature modification reduces feature suppression, enhancing the representation of both MNIST and STL10 features simultaneously. All IFM runs use a fixed value $\varepsilon = 0.1$, and loss $\mathcal{L} + 0.5 \cdot \mathcal{L}_\varepsilon$ (i.e. weighting parameter $\alpha = 0.5$) to illustrate robustness to the choice of parameters.

for adversarial finetuning. We perform experiments on CIFAR10 and CIFAR100 and the results are shown in Fig. 5.

**Results** See Fig. 5 in the main manuscript for the results. There are significant qualitative differences between the behaviour of IFM and ACL(DS). IFM improves (standard) linear readout accuracy with zero memory or compute time cost increase, whereas ACL(DS) has improved adversarial linear readout performance, but at the cost of worse standard linear readout and $2\times$ memory and $6\times$ time per epoch. This shows that these two method are addressing two distinct problems. ACL(DS) is suitable for improving the adversarial robustness of a model, whereas IFM improves the generalization of a representation.

## C.4 Object classification experiments

We first describe the protocol used for evaluating IFM on the following datasets: CIFAR10, CIFAR100, STL10, tinyImageNet. For simplicity, the objective weighting parameter is fixed at $\alpha = 1$. For MoCo-v2, we performed 5-fold cross validation for CIFAR10/CIFAR100 datasets, and 3 replicated runs on official train/val data splits for tinyImageNet and STL10 datasets.

**Training** All encoders have ResNet-50 backbones and are trained for 400 epochs with temperature $\tau = 0.5$ for SimCLR and $\tau = 0.1$ for MoCo-v2. Encoded features have dimension 2048 and are followed by a two layer MLP projection head with output dimension 128. Batch size is taken to be 256, yielding negative batches of size $m = 510$ for SimCLR. For MoCo-v2, we use a queue size of $k = 4096$ (except for STL10 dataset we use $k = 8192$), and we use batch size of 256 for CIFAR10, CIFAR100 and tinyImageNet, 128 for STL10. For both SimCLR and MoCo-v2 we use the Adam optimizer.

SimCLR uses initial learning rate $1 \times 10^{-3}$ and weight decay $1 \times 10^{-6}$ for CIFAR10, CIFAR100 and tinyImageNet, while STL10 uses $1 \times 10^{-1}$ learning rate, and weight decay $5 \times 10^{-4}$ (since we found these settings boosted performance by around $5\%$ in absolute terms). MoCo-v2 training uses weight decay $5 \times 10^{-4}$, and an initial learning rate $3 \times 10^{-2}$ for CIFAR10 and CIFAR100; and learning rate $1 \times 10^{-1}$ for STL10 and tinyImageNet. Cosine learning rate schedule is used for MoCo-v2.

**Linear evaluation** Evaluation uses test performance of a linear classifier trained ontop of the learned embedding (with embedding model parameters kept fixed) trained for 100 epochs.

For SimCLR, the batch size is set as 512, and the linear classifier is trained using the Adam optimizer with learning rate $1 \times 10^{-3}$ and weight decay $1 \times 10^{-6}$, and default PyTorch settings for other hyperparameters. For CIFAR10 and CIFAR100 the same augmentations as SimCLR are used for linear classifier training, while for STL10 and tinyImageNet no augmentations were used (since we found this improves performance).

For MoCo-v2, the batch size is set as 256. Training uses SGD with initial learning rate set to 30, momentum is set as 0.9 and a scheduler that reduces the learning rate by a factor of $10\%$ at epoch 30 and 60. The weight decay is 0. For CIFAR10 and CIFAR100, we normalize images with mean of $[0.4914, 0.4822, 0.4465]$ and standard deviation of $[0.2023, 0.1994, 0.2010]$. For STL10 and tinyImageNet, we normalize images with mean of $[0.485, 0.456, 0.406]$ and standard deviation of $[0.229, 0.224, 0.225]$. The same augmentations as the official MoCo-v2 implementation are used for linear classifier training.

### C.4.1 ImageNet100

We adopt the official MoCo-v2 code[2] (CC-BY-NC 4.0 license), modifying only the loss function. For comparison with AdCo method, we adopt the official code[3] (MIT license) and use the exact same hyperparmeters as for MoCo-v2. For the AdCo specific parameters we perform a simple grid search for the following two hyperparameters: negatives learning rate $lr_{\text{neg}}$ and negatives temperature $\tau_{\text{neg}}$. We search over all combinations $lr_{\text{neg}} \in \{1, 2, 3, 4\}$ and $\tau_{\text{neg}} \in \{0.02, 0.1\}$, which includes the AdCo default ImageNet1K recommendations $lr_{\text{neg}} = 3$ and $\tau_{\text{neg}} = 0.02$ [20]. The result reported for AdCo in Tab. 1 is the best performance over all 8 runs.

**Training** We use ResNet-50 backbones, and train for 200 epochs. We use a base learning rate of $0.8$ with cosine annealing scheduling and batch size 512. The MoCo momentum is set to $0.99$, and temperature to $\tau = 0.2$. All other hyperparameters are kept the same as the official defaults.

**Linear evaluation** We train for 60 epochs with batch size 128. We use initial learning rate of $30.0$ and a step scheduler that decreases the learning rate by a factor of 10 after epochs: $30, 40, 50$. All other hyperparameters are kept the same as the official MoCo-v2 defaults.

As noted in the manuscript, our combination of training and linear evaluation parameters leads to $80.5\%$ top-1 linear readout for standard MoCo-v2, and $81.4\%$ with IFM-MoCo-v2. The standard MoCo-v2 performance of $80.5\%$ is, to the best of our knowledge, state-of-the-art performance on ImageNet100 using 200 epoch training with MoCo-v2. For comparison, we found that using the default recommended MoCo-v2 ImageNet1k parameters (both training and linear evaluation) achieves ImageNet100 performance of $71.8\%$. This choice of parameters maybe useful for other researchers using MoCo-v2 as a baseline on ImageNet100.

### C.5 COPDGene dataset

The dataset [38] in our experiments includes 9,180 subjects. Each subject has a high-resolution inspiratory CT scan and five COPD related outcomes, including two continuous spirometry measures: (1) FEV1pp: the forced expiratory volume in one second, (2) $FEV_1/FVC$: the FEV1pp and forced vital capacity (FVC) ratio, and three ordinal variables: (1) six-grade centrilobular emphysema (CLE) visual score, (2) three-grade paraseptal emphysema (Para-septal) visual score, (3) five-grade dyspnea symptom (mMRC) scale. The dataset is publicly available.

For fair comparison, we use the same encoder and data augmentation described in the baseline approach [42]. We set the representation dimension to 128 in all experiments. For simplicity, instead of using a GNN, we use average pooling to aggregate patch representations into image representation. The learning rate is set as $0.01$. We use Adam optimizer and set momentum as $0.9$ and weight decay as $1 \times 10^{-4}$. The batch size is set as 128, and the model is trained for 10 epochs.

### C.6 Further discussion of feature robustness experiments (Sec. 4.3)

Ilyas et al. [22] showed that deep networks richly represent so-called "non-robust" features, but that adversarial training can be used to avoid extracting non-robust features at a modest cost to downstream performance. Although in-distribution performance is harmed, Ilyas et al. argue that the reduction in use of non-robust features – which are highly likely to be statistical coincidences due to the high dimensionality of input data in computer vision – may be desirable from the point of view of trustworthiness of a model under input distribution shifts. In this section we consider similar questions on the effect implicit feature modification on learning of robust vs. non-robust features during self-supervised pre-training.

Compared to supervised adversarial training [22, 30] our approach has the key conceptual difference of being applied in feature space. As well as improved computation efficiency (no PGD attacks required) Fig. 8 shows that this difference translates into different behavior when using implicit feature modification. Instead of suppressing non-robust features as Ilyas et al. observe for supervised representations, IFM *enhances the representation of robust features*. This suggests that the improved generalization of encoders trained with IFM can be attributed to improved extraction of features aligned with human semantics (robust features). However, we also note that IFM has no significant

---

[2]https://github.com/facebookresearch/moco
[3]https://github.com/maple-research-lab/AdCo

effect on learning of non-robust features. In Appdx. D we discuss the idea of combining IFM with adversarial training methods to get the best of both worlds.

# D   Discussion of limitations and possible extensions

While our work makes progress towards understanding, and controlling, feature learning in contrastive self-supervised learning, there still remain many open problems and questions. First, since our proposed implicit feature modification method acts on embedded points instead of raw input data it is not well suited to improving $\ell_p$ robustness, and similarly is not suited to removing pixel-level shortcut solutions. Instead our method focuses on high-level semantic features. It would be valuable to study the properties of our high-level method used in conjunction with existing pixel-level methods.

Second, while we show that our proposed implicit feature modification method is successful in improving the representation of multiple features simultaneously, our method does not admit an immediate method for determining *which* features are removed during our modification step (i.e. which features are currently being used to solve the instance discrimination task). One option is to manually study examples using the visualization technique we propose in Sec. 3.1.