# OpenReview forum: "Can contrastive learning avoid shortcut solutions?"
_NeurIPS.cc/2021/Conference — NeurIPS 2021 Poster_

### Official Review · Reviewer_yWao · 2021-07-05

**Rating:** 6
**Confidence:** 3

**Summary:**

This paper studies how to avoid shortcut solutions in contrastive learning.  To this end, this work explores feature suppression in contrastive learning and explains why shortcut solutions happen. Then, this work proposes a simple method to handle this issue based on the observation on the difficulty of instance discriminations. Experimental results show the effectiveness of the proposed method.

**Limitations And Societal Impact:**

Please refer to the above main review.

**Main Review:**

Positive points:
1. This paper studies an important and practical issue in contrastive learning: how to avoid shortcut solutions.
2. This paper theoretically analyzes feature suppression in contrastive learning and explains why it may happen in contrastive self-supervised learning.
3. This paper proposed a simple and efficient method based on adjusting the difficulty of instance discrimination.

Negative points:
1. The notations are two complex, which makes the paper hard to follow. Moreover, some notations seem to not be defined well. For example, what is $\bar{z}$ in Definition 1? What is $S^c$ in Line 194.

2. It is would be better to conduct experiments on ImageNet-1k. All advanced contrastive learning methods are evaluated on ImageNET-1k, which can better evaluate the effectiveness of the proposed method.

3. Whether does the proposed method contribute to siamese architectures, like BYOL and Simsiam?

Typos: "an" seems "and" in Line 172.

**Time Spent Reviewing:**

7

---

> ### Author Response · Authors · 2021-08-09
> **Thank you for your encouraging review**
>
> Thank you for your encouraging review, and useful comments. We would like to discuss the points raised in your review.
>
> ---
>
> **Notational clarity**
>
> Since the submission in May we have revised and simplified the notation we use in order to make the theoretical parts of the paper more accessible. On the two specific questions you asked:
> - what is $\bar{z}$ in Definition 1?: we let $z$ and $\bar{z}$ denote any two points in our latent space $\mathcal{Z}$, so $\bar{z}$ plays a symmetric role to the variable $z$.
>
> - What is $S^c$ in l.194?: $S$ is a subset of the discrete set $[n]=${$0,1,2, \ldots, n$}. We use $S^c$ to denote the complement of $S$ in \{$0,1,2, \ldots, n$}.
>
> We will update the paper to clarify both of these points.
>
> ---
> **Adapting IFM to Siamese methods (e.g. SimSiam and BYOL)**
>
> Our method could be adapted to Siamese methods, but with one additional complication. IFM is defined via a maximization problem over perturbations in embedding space. With SiamSiam, for example, these perturbations would need to be computed *after* the projection head, but *before* the predictor MLP. Computation would require using gradient accent with backpropagation through the predictor MLP. Since the predictor MLP is small this would be relatively inexpensive, but would still add some computational overhead. This approach also lacks the analytic expression for the IFM loss (Eqn. 2) in the contrastive learning setup. In this work we decided to focus on contrastive methods, since much of our motivation for IFM comes from Section 2, which considers properties (e.g. hardness) of the *instance discrimination* task. Since Siamese methods do not use instance discmriniation, the observations in Section 2 do not directly transfer. We agree, however, that analyzing feature learning in Siamese methods is a valuable direction for future work.
>
> ---
>
> **ImageNet-1k experiments**
>
> We would very much like to run ImageNet-1k experiments. However, computational resource constraints make this very difficult for us. We are attempting to run these experiments, but they are extremely time-consuming (on the order of many weeks to train). If we are able to complete these experiments we look forward to including them.
>
> We welcome any further questions, and are very happy to discuss further.

---

> > ### Comment · Reviewer_yWao · 2021-08-12
> > **Thanks for response**
> >
> > Thanks for your response. I am happy to see the discussion of adapting to Siamese methods, which will make the paper more comprehensive.
> >
> > In addition, I fully understand that it may take more time to obtain the result on ImageNet-1K due to computational resource constraints, but I am still expecting to see such results. Considering all advanced contrastive learning methods are evaluated on ImageNET-1k, adding such results will further improve the quality of this paper. If the performance is promising, I am happy to raise my rating.

---

### Official Review · Reviewer_hQmV · 2021-07-16

**Rating:** 7
**Confidence:** 3

**Summary:**

This paper provides a theoretical study on the cause of feature suppression in contrastive learning, and proposes "implicit feature modification" which is a theoretically grounded, yet simple and effective way of improving (avoiding) feature suppression in contrastive learning.

**Limitations And Societal Impact:**

NA.

**Main Review:**

**Pros:**

1. Interesting theoretical analysis to explain the feature suppression phenomenon in contrastive learning.
2. Well-designed experiments to demonstrate claims made in the paper, e.g. more difficult instance discrimination task makes the model learn more difficult features but at the same time suppressing the original learned features.
3. Simplicity of IFM. No computational overheads, theoretically supported and shown to be effective on a decent variety of datasets. It is also nice to see IFM helps learn robust features.

**Cons:**

The approach is not tested on larger scale datasets such as ImageNet-1k, probably due to computational constraints. I'm curious about the impact of the approach on larger datasets, where diversity of features and amount of images may naturally help contrastive learning to learn more features. The impact of IFM would be even more if it is shown to be effective for even larger datasets.

**Time Spent Reviewing:**

3

---

> ### Author Response · Authors · 2021-08-08
> **Thank you for the positive review**
>
> Thank you for the positive review and encouraging comments. We are glad that you appreciated the various components of this work, including theory supported by experiments, whose new insights motivated the design of a simple and efficient method for reducing the tendency to learn shortcuts.
>
> **ImageNet-1K experiments:**  we would very much like to run these experiments. As you suggested, unfortunately computational resource constraints make this very difficult for us. We are attempting to run these experiments now, but they are extremely time-consuming (on the order of many weeks to train). If we are able to complete these experiments we are eager to see the results and will include them in an updated version of the paper.
>
> We are very happy to address any further questions you may have.

---

> > ### Comment · Reviewer_hQmV · 2021-08-30
> > **Response**
> >
> > Thank you for your response. I will keep my score and still recommend for acceptance.

---

### Official Review · Reviewer_c2ts · 2021-07-17

**Rating:** 8
**Confidence:** 3

**Summary:**

In this paper, the authors assess whether contrastive instance discrimination can be modified to avoid models that learn “shortcut” solutions and discard important features.  They use a theoretical approach to explain why optimizing the InfoNCE loss that is associated with instance discrimination does not necessarily avoid shortcut solutions, and demonstrate that it is possible to adjust the difficulty of instance discrimination in a relatively straightforward way to tune the tradeoff between which features are learned.  The authors propose a method called Implicit Feature Modification (IFM) which encourages encoders to use multiple different input features to discriminate between instances in a cost-efficient manner that improves generalization and reduces suppression of useful features.  They provide a variety of empirical results supporting both the reasoning behind and the practical significance of their approach.

**Ethical Concerns:**

None.

**Limitations And Societal Impact:**

The authors discuss these issues in their Appendix.  One potential addition would be a discussion and/or analysis of if this approach yields different levels of performance gains (or potential losses) on some classes versus others, rather than an analysis based purely on aggregate performance metrics.  Though this addition would be valuable, the authors’ existing discussion is informative.

**Main Review:**

Originality:

While I am not the most familiar with all of the related work in this area, the approach appears novel and the differences between the current work and cited work is clearly explained.  The narrative arc taking the reader from analysis of feature suppression through representation tradeoffs to motivating IFM is compelling and the arguments are well-made.


Quality:

The submission appears technically sound.  The authors use appropriate methods to make their arguments, provide experimental justification for the claims the paper makes, and provide detailed descriptions of their experimental protocols.  The combination of theoretical analysis and empirical results that combine to tell a clear story is compelling.  The authors have an explicit limitations and broader impact section that adequately evaluates the strengths and weaknesses of their work.  I would, however, appreciate an additional discussion of the computational cost of this approach relative to the overall training process, as this seems a critical component of the work’s practical applicability that I do not remember seeing in the paper.

Clarity:

I found the paper to be extremely well-written.  However, some of the mathematical exposition is notationally dense, and it may be possible to make the paper more accessible by simplifying the presentation.

Significance:

As a practitioner in computer vision, I find these results important and would certainly expect others to use these ideas.  The fundamental problem of models that take shortcuts to achieve high levels of performance on a training set, but perform poorly in practice, is extremely common and often addressed only in ad hoc ways.  The authors present one of the first methods of which I am aware by which this issue can be systematically addressed, they motivate/analyze it theoretically, and provide compelling empirical results to support its practical significance.  The fact that the authors show improvement not only on benchmark datasets, but also on a medical imaging dataset that reflects a real application, is particularly compelling.

Detailed Comments/Questions:

L 71-75: I very much appreciate the authors’ presentation of this point.

L 92: What is the “space U” in “measure \nu on a space U”?

L 83 - 95: It may be worth describing why the authors set up the problem in this way.  It is not immediately obvious what the benefit of this formulation is at this juncture.

L 112: Nit: “number of negatives m goes to infinity” → as the reader, I did not know what m was without reading the following lines.

L112: It’s not immediately clear to me how the LFS → RHS.  Perhaps worth working out in an appendix or citing directly.

L 133 - 135: Another explanatory sentence here may be useful to provide intuition.

L 229: Do you evaluate sensitivity to alpha at any point, or is alpha=1 generally a good choice?
Fig. 4 caption: Isn’t this moving the negative sample towards the anchor (rather than away, as the caption says)?

L 294: It seems that the difference in linear readout accuracy b/t IFM and standard SIMCLR is very small (if not overlapping) in many cases.  I would soften the statement that it improves linear readout accuracy across all three features for all temperature settings.

Fig. 7: What does L_epsilon only mean?  Is this just not using the standard InfoNCE loss at all?  Why is this better sometimes for MoCo-v2?  Potentially worth discussing.

Table 1: Please show confidence intervals here or some measure of spread.  Point estimates are not sufficient.

L 328: Typo in “previouos”

L 337: This should be Appendix C.5 not C.6, I think.

Table 2: Some of the bolded numbers are likely not statistically significant (e.g. mMRC column).  Please indicate exactly what bold means, and ensure that it accurately reflects statistical significance in some sense.

Discussion:  Extremely short for a section entitled “discussion.”  I would suggest calling this “conclusions” and adding a bit more material here to summarize the results presented.


**Time Spent Reviewing:**

5

---

> ### Author Response · Authors · 2021-08-09
> **Thank you for your kind words and encouraging review**
>
> We are glad that you value to practical implications of our work, and its contribution to understanding the problem of shortcuts. We also especially appreciate your effort in compiling a list of useful actionable improvements to the paper. We will incorporate all of them into the paper, and they will improve the paper and its readability further.
>
> Besides these useful suggestions, your review also raises several questions, which we clarify next.
>
> ---
>
> **The computational cost of IFM:** IFM introduces (essentially) zero computational overhead. Figure 5 shows a comparison of both memory and runtime between SimCLR and IFM demonstrating this. The reason is that, although defined via a maximization problem, IFM can be computed analytically using a simple perturbation of the logits (as shown in Eqn. 2).
>
>
> **The space $\mathcal{U}$ in l.92:** $\mathcal{U}$ is any measurable space. We use this to define the pushforward measure in full generality (which is defined for any two measurable spaces $\mathcal{U}, \mathcal{V}$ and any $f:\mathcal{U} \rightarrow \mathcal{V}$). In our theoretical analysis, the particular instance of the pushforward we use is with $\mathcal{U}$ equal to the unit sphere $\mathbb{S}^{d-1} =$ {$ x \in \mathbb{R}^d : ||x||_2 = 1$}.
>
> **What is $m$ in “number of negatives $m$ goes to infinity”, l.112?:** We define $m$ in Eqn (1). It is the size of the batch of negatives used in the InfoNCE loss. We will add further clarification to l.112 so the reader does not have to refer back to (1).
>
> **The limit LFS → RHS in l.112**: We will add the derivation in the appendix, and refer the reader to the details if they are interested.
>
> **The sensitivity to $\alpha$ (the weighting between the InfoNCE and the perturbed loss):** We did not tune $\alpha$. When designing our experiments we decided to value simplicity more highly than attaining optimal performance. It may be possible to further improve the performance of IFM by tuning alpha. However, our experiments suggest that indeed $\alpha=1$ is a generally sensible choice. We are happy to add an ablation in the appendix where we vary $\alpha$.
>
> **What does "$\mathcal{L}_\epsilon$ only" mean in Fig. 7?:** It means that  instead of the default IFM loss, which is the sum with the standard InfoNCE loss, $(\mathcal{L}_\epsilon +\mathcal{L})/2$, we only optimized the loss $\mathcal{L}_\epsilon$. We will make sure to clarify this in an updated version of the paper. We found that $\mathcal{L}_\epsilon$ performs similarly to $\mathcal{L}$ on average (76.0% vs 75.9% on average across all 8 runs). We will add confidence intervals that should further clarify this.
>
> **Confidence intervals for Table 1:** We agree this valuable and are running these experiments now. We will add the results to the paper. Unfortunately, due to computational constraints they have not yet finished.
>
> **Statistical significance for Table 2:** The bold face in Table 2 indicates the best average performance in 5-fold cross-validation. We will clarify this in text and update the figure to carefully denote account statistical significance.
>
> ---
>
> Finally, thank you for pointing out typos, incorrect appendix references (C.5 vs C.6) and points on style (e.g. “discussion” vs “conclusion''). We will also incorporate these into the paper.

---

> > ### Comment · Reviewer_c2ts · 2021-08-30
> > **Appreciate the Response**
> >
> > Dear Authors,
> >
> > First, thank you for taking the time to respond in detail.  Several responses:
> >
> > (1) Re: performance, the bottom panels of Fig. 5 do indeed provide the information I was looking for.   I might recommend making this figure slightly larger, as it was hard to read.
> >
> > (2) I appreciate the clarification here re: U.  More broadly speaking, as one of the other reviewers also pointed out, generally decreasing the complexity of the notation would be a positive thing.
> >
> > (3) I just looked at Eq. (1), and see that m is in the equation, but it still was not immediately obvious to me given that it was never defined in the text (unless I missed it again).  In any case, I would encourage the authors to define m explicitly in the text to avoid confusion.
> >
> > (4) I appreciate the authors providing the confidence intervals.  While I agree that the CIs presented do appear to indicate statistical significance, I would also emphasize that the absolute improvements may be modest in some settings.
> >
> > (5) Re: statistical significance in Table 2, yes, please add detail indicating that bold indicates best performance.  More to the point, do *not* bold items where the result is *not* statistically significant, as in these cases one cannot reliably claim that one method is better than another.  E.g. in the mMRC column of the original paper, the bold appears to be misleading.  I would further suggest adding to your discussion that not all columns in Table 2 yield a statistically significant improvement using IFM (assuming, as appears to be the case to me, this is true).
> >
> > I have maintained my rating with the expectation that the above points are resolved in the final draft.

---

> > > ### Author Response · Authors · 2021-08-30
> > > **thank you for your review!**
> > >
> > > Thank you for confirming your rating. We are adding / have added each of these points to the manuscript, and it will be fully completed for a final draft. We are grateful for your suggestions which are actionable, and will help improve the paper and it's clarity.
> > >
> > > Best wishes

---

### Official Review · Reviewer_5txB · 2021-07-25

**Rating:** 6
**Confidence:** 4

**Summary:**

This paper investigates feature learning in contrastive learning. It argues that feature suppression occurs in contrastive learning, where the model ignores a subset of the features. It argues that by varying the difficulty of the contrastive discrimination task, it is possible to incentivize the learning of particular features, but again at the expense of other features. To deal with this feature suppression problem, the authors propose a new method called implicit feature modification, where the basic idea is to continuously remove features from the embedding representation that are being used to discriminate negatives from positives. This encourages the model to learn new, additional features to perform the discrimination task. The authors show that this scheme makes the model learn more robust features (in the sense of Ilyas et al. (2019) --ref. 22--).

**Limitations And Societal Impact:**

I mentioned my views on the limitations of this paper above. It would be great if the authors could address my concerns about the significance of the effects and cross-validation. I also think that in their experiments, the authors could use stronger baselines to convince the readers that their sophisticated method is really needed for the reported improvements. For example, could simply adding a drop-out layer (or a similar regularizing layer) before the embedding also encourage the model to use more features and lead to similar improvements?

**Main Review:**

I think this is a technically sound, novel contribution to the contrastive learning literature. I think the proposed method conceptually makes sense. Some of the experiments seem to be pretty well done. However, some experiments do not have any error bars (e.g. Figure 7-8, Table 1). Are those effects significant? Also, the reported improvements are rather small and the need to tune an additional hyperparameter (epsilon) makes the proposed scheme practically unappealing. I also want to point out that the authors never do a proper cross-validated experiment: they just show the results for different epsilon values (e.g. Figure 7, Table 2), which again makes me wonder if these effects would survive a properly cross-validated experiment.

I also had a more high level objection that I wanted to mention. This is more of an objection to contrastive learning in general, rather than this particular paper, but the point is that what papers like this really show for me is that instance discrimination and similar objectives as self-supervised learning objectives are fundamentally flawed (although they happen to work quite well at the scales where we're currently testing them), because there’s no guarantee that they will learn the “intended” representation. These objectives already require quite a bit of manual data augmentation engineering to cajole the model into learning the “correct” representation and papers like this suggest that even then these methods are not entirely successful at preventing shortcut learning. So, instead of trying to patch these issues with ad-hoc fixes like this paper tries to do, I think it would be more productive over the long term to adopt more principled self-supervised learning objectives in the first place (for example, likelihood-based generative objectives), where at least in the limit of large data, we know that they will learn the right features.

**Time Spent Reviewing:**

4

---

> ### Author Response · Authors · 2021-08-08
> **Thank you for your suggestions - we are running cross-validation now**
>
> Thank you for your review, and encouraging comments. We are glad that you appreciated the various components of this work, including technical soundness, novelty, and execution of experiments.
>
> Your review raised a number of points, which we discuss below.
>
> ---
>
> **Cross-validation and error bars**
>
> We fully agree that this is a valuable addition to the experimental results and **will add 5-fold cross-validation to all remaining plots in an updated version of the paper** (for STL10 and ImageNet100 datasets we will instead run multiple seeds and add error bars).
>
> Due to computational constraints, we have partial results so far. The results reported below use 5-fold cross-validation, and report 95% confidence intervals.
>
> ```
> Dataset: CIFAR100
>
>       Method       | epsilon | avg. accuracy | 95% CI |
> -------------------------------------------------------
> MoCo-v2            |   N/A   |     68.31%    | ±0.61% |
> MoCo-v2 w. dropout |   N/A   |     68.16%    | ±0.78% |
> IFM-MoCo-v2        |   0.1   |     69.80%    | ±0.60% |
> ```
>
> ```
> Dataset: CIFAR10
>
>    Method   | epsilon | avg. accuracy | 95% CI |
> -------------------------------------------------------
> MoCo-v2            |   N/A   |     91.69%    | ±0.25% |
> MoCo-v2 w. dropout |   N/A   |     91.84%    | ±0.13% |
> IFM-MoCo-v2        |   0.1   |     92.28%    | ±0.20% |
> ```
>
> ```
> Dataset: tinyImageNet
>
>    Method   | epsilon | avg. accuracy | 95% CI |
> ------------------------------------------------
> MoCo-v2     |   N/A   |     51.78%    | ±0.50% |
> IFM-MoCo-v2 |   0.1   |     52.41%    | ±0.52% |
> ```
> ```
> Dataset: ImageNet-100
>
>    Method   | epsilon | avg. accuracy | 95% CI |
> ------------------------------------------------
> MoCo-v2     |   N/A   |     80.45%    | ±0.11% |
> IFM-MoCo-v2 |   0.1   |     80.91%    | ±0.25% |
> ```
>
> These results show that IFM also yields statistically significant improvements when using cross-validation. We will also follow up at the end of the rebuttal period if further results have finished running by then.
>
> While we continue to prepare additional results, we would also like to emphasize that across all 5 datasets considered and with both SimCLR and MoCo-v2 frameworks, each $\epsilon$ in the set {$0.05, 0.1, 0.2$} improved performance over the baseline (and so did $\epsilon=0.5$ in 7 out of 8 cases). The consistency of this observation across different experimental settings gives us high confidence in the overall conclusion that IFM improves over the baseline.
>
> ---
>
> **Other ways of reducing feature suppression**
>
> It is possible that methods such as dropout may also help with feature suppression. However, we view the value of IFM as coming from its motivation via the observations in Section 2 on the relation between shortcut learning and instance discrimination. That IFM is motivated in this way serves two purposes: 1) it shows that the understanding developed in Section 2 can help inspire new methodology, and 2) conversely, the effectiveness of IFM further validates the viewpoint developed in Section 2.
>
> ---
>
> **Tuning the parameter $\epsilon$**
>
> We are able to offer a sensible default setting: $\epsilon=0.1$. As noted above, all epsilon values {$0.05, 0.1, 0.2$} improved performance on all datasets, however $\epsilon=0.1$ marginally outperformed $0.05$ and $0.2$ on average. That all three of the settings {$0.05, 0.1, 0.2$}  yielded improvements suggests that $\epsilon$ is not a sensitive hyperparameter.
>
> ---
>
> **Fundamental limitations of instance discrimination**
>
> We fully agree that contrastive instance discrimination is intrinsically difficult to guide towards learning the “correct” representation (as are all augmentation-based self-supervised methods, e.g. BYOL, Barlow Twins, SwAV). Indeed this is one of the takeaways from our work. We think this problem is one of the major open questions in representation learning, and intersects with many other lines of research including distributional robustness, domain adaptation, out-of-distribution generalization, implicit biases, and more.
>
> Although this criticism is not specific to our work (as you mentioned), we would still like to offer two reasons why IFM is a valuable contribution with this open challenge in mind. First, in the absence of more principled methods (which may take years to develop), it is of interest to build new tools that can be immediately deployed by practitioners. Second -- and we think this point is especially important -- the efficacy of IFM serves as a final validation of the observations in Section 2 (which motivated IFM). By offering a better understanding of instance discrimination’s idiosyncrasies, IFM helps clarify its limitations and may motivate further developments.
>
> We think this is an interesting discussion and would be happy to add it to the paper, should you wish.

---

> > ### Author Response · Authors · 2021-08-10
> > **dropout baseline added**
> >
> > We would like to leave a brief note to notify you that, as suggested, we added results to the figures above where we trained models using a drop-out layer before the embedding to determine if this approach would act similarly to IFM. On CIFAR10/100 we find that the dropout layer does not match IFM.

---

> > > ### Comment · Reviewer_5txB · 2021-08-27
> > > **thank you**
> > >
> > > I thank the authors for running these additional experiments and analyses. I'm happy to increase my score as a result. As I mentioned in my initial review, the improvements seem a bit small and I still maintain my overall concern about contrastive learning in general. However, I do think this paper offers some valuable insights on the workings of contrastive learning.

---

> > > > ### Author Response · Authors · 2021-08-29
> > > > **Thank you for your review**
> > > >
> > > > Thank you for the positive feedback! We will continue working on adding the rest of the experiments.
> > > >
> > > > Best wishes

---

### Decision · Program_Chairs · 2021-09-27

**Decision:**

Accept (Poster)

**Comment:**

The reviewers agreed that this paper addresses and important practical question (feature supression in contrastive learning), and provides a theoretically-motivated and empirically-validated way to address it.